# EXPLORING THE GENERALIZATION CAPABILITIES OF AID-BASED BI-LEVEL OPTIMIZATION

## ABSTRACT

Bi-level optimization has achieved considerable success in contemporary machine learning applications, especially for given proper hyperparameters. However, due to the two-level optimization structure, commonly, researchers focus on two types of bi-level optimization methods: approximate implicit differentiation (AID)-based and iterative differentiation (ITD)-based approaches. ITD-based methods can be readily transformed into single-level optimization problems, facilitating the study of their generalization capabilities. In contrast, AID-based methods cannot be easily transformed similarly but must stay in the two-level structure, leaving their generalization properties enigmatic. In this paper, although the outer-level function is nonconvex, we ascertain the uniform stability of AID-based methods, which achieves similar results to a single-level nonconvex problem. We conduct a convergence analysis for a carefully chosen step size to maintain stability. Combining the convergence and stability results, we give the generalization ability of AID-based bi-level optimization methods. Furthermore, we carry out an ablation study of the parameters and assess the performance of these methods on real-world tasks. Our experimental results corroborate the theoretical findings, demonstrating the effectiveness and potential applications of these methods.

## 1 INTRODUCTION

As machine learning continues to evolve rapidly, the complexity of tasks assigned to machines has increased significantly. Thus, formulating machine learning tasks as simple minimization problems is not enough for complex tasks. This scenario is particularly evident in the scenarios of meta-learning and transfer learning tasks. To effectively tackle these intricate tasks, researchers have turned to the formulation of problems as bi-level formulas. Conceptually, this can be represented as follows:

$$\min_{x \in \mathbb{R}^{d_x}, y^*(x) \in \mathbb{R}^{d_y}} \left\{ \frac{1}{n} \sum_{i=1}^{n} f(x, y^*(x), \xi_i), \ \ s.t. \ \ y^*(x) \in \arg\min_{y \in \mathbb{R}^{d_y}} \frac{1}{q} \sum_{j=1}^{q} g(x, y, \zeta_j) \right\}, \quad (1)$$

where $d_x$ and $d_y$ are the dimensions of variables x and y, respectively. $\xi_i$ represents samples from $D_v \in \mathcal{Z}_v^n$, while $\zeta_j$ are samples from $D_t \in \mathcal{Z}_t^q$, where $\mathcal{Z}_v$ and $\mathcal{Z}_t$ are the sample space of the upper-level problem and the lower-lever problem, respectively. Functions $f$ and $g$ are nonconvex yet smooth, with $f$ applying to both $x$ and $y$, while $g$ is strongly convex and smooth for $y$.

Consider the example of hyper-parameter tuning. In this context, $x$ is treated as the hyper-parameters, while $y$ represents the model parameters. The optimal model parameters under the training set $D_t$ can be expressed as $y^*(x)$ when a hyperparameter $x$ is given. The performance of these parameters is then evaluated on the validation set $D_v$. Yet, in practice, gathering validation data can be costly, leading to the crucial question of the solution's generalizability from the validation set to real scenarios.

The solutions to such bi-level optimization problems in the machine learning community have conventionally relied on two popular methods: Approximate Implicit Differentiation (AID)-based methods and Iterative Differentiation (ITD)-based methods. While ITD-based methods are intuitive and easy to implement, they are memory-intensive due to their dependency on the optimization trajectory of $y$. AID-based methods, on the other hand, are more memory-efficient.

Recently, Bao et al. (2021) have proposed a uniform stability framework that quantifies the maximum difference between the performance on the validation set and test set for bi-level formulas, which belongs to ITD-based methods. For ITD-based methods, the trajectory of $y$ can be easily written as a

function of current iterates $x$ making it easy to be analyzed as a single-level optimization method. However, for AID-based methods, a similar analysis is complex due to the dependence of the current iterates $x$ and $y$ on previous ones, making generalization a challenge.

In this paper, we focus on studying the uniform stability framework for AID-based methods. We present a stability analysis for non-convex optimization with various learning rate configurations. A noteworthy finding is that when the learning rate is set to $\mathcal{O}(1/t)$, we can attain results analogous to those in single-loop nonconvex optimization. Furthermore, we present convergence results for AID-based methods and highlight the trade-off between optimization error and generalization gaps.

In summary, our main contributions are as follows:

- We have developed a novel analysis framework aimed at examining multi-level variables within the stability of bi-level optimization. This framework provides a structured methodology to examine the behavior of these multi-level variables.
- Our study reveals the uniform stability of AID-based methods under a set of mild conditions. Notably, the stability bounds we've determined are analogous to those found in nonconvex single-level optimization and ITD-based bi-level methods. This finding is significant as it supports the reliability of AID-based methods.
- By integrating convergence analysis into our research, we've been able to unveil the generalization gap results for certain optimization errors. These findings enhance our understanding of the trade-offs between approximation and optimization in the learning algorithms. Furthermore, they provide practical guidance on how to manage and minimize these gaps, thereby improving the efficiency and effectiveness of bi-level optimization methods.

## 2   RELATED WORK

**Bilevel Optimization.** Franceschi et al. (2017; 2018) use bilevel optimization to solve the hyperparameter problem. Besides, Finn et al. (2017) and Rajeswaran et al. (2019) leverage bilevel optimization to solve the few-shot meta-learning problem. Besides the above research areas, researchers also apply bi-level to solve neural architecture search problems. Liu et al. (2018), Jenni and Favaro (2018), and Dong et al. (2020) all demonstrate the effectiveness of bilevel optimization for this task. Additionally, bilevel optimization can be used to solve min-max problems, which arise in adversarial training. Li et al. (2018) and Pfau and Vinyals (2016) use bilevel optimization to improve the robustness of neural networks. Moreover, researchers explore the use of bilevel optimization for reinforcement learning. Pfau and Vinyals (2016) and Wang et al. (2020) use bilevel optimization to improve the efficiency and effectiveness of reinforcement learning algorithms. In addition, Ghadimi and Wang (2018), Hong et al. (2020), Dagréou et al. (2022), Tarzanagh et al. (2022) and Chen et al. (2022) show the convergence of various types of bi-level optimization methods under stochastic, finite-sum, higher-order smoothness, federated learning, and decentralized settings, respectively.

**Stability and Generalization Analysis.** Bousquet and Elisseeff (2002) propose that by changing one data point in the training set, one can show the generalization bound of a learning algorithm. They define the different performances of an algorithm when changing the training set as stability. Later on, people extend the definition in various settings, Elisseeff et al. (2005) and Hardt et al. (2016) extend the algorithm from deterministic algorithms to stochastic algorithms. Hardt et al. (2016) gives an expected upper bound instead of a uniform upper bound. Chen et al. (2018) derive minimax lower bounds for single-level minimization tasks. Ozdaglar et al. (2022) and Xiao et al. (2022) consider the generalization metric of minimax setting, and Bao et al. (2021) extend the stability to bi-level settings. Different from the previous works, as far as we know, we are the first work that gives stability analysis for AID-based bi-level optimization methods.

## 3   PRELIMINARY

In this section, we explore two distinct types of algorithms: the AID-based algorithm (referenced as Algorithm 1) and the ITD-based algorithm (referenced as Algorithm 2). Further, we will give the decomposition of generalization error for bi-level problems.

### 3.1   BI-LEVEL OPTIMIZATION ALGORITHMS

Before delving into the detailed operation of the AID-based methods, it is crucial to comprehend the underlying proposition that governs its update rules. Let us define a function $\Phi(x) = \frac{1}{n} \sum_{i=1}^{n} f(x, y^*(x), \xi_i)$. This function has the gradient property as stated below:

---

**Algorithm 1** AID Bi-level Optimization Algorithm

---
1: Initialize $x_0, y_0, m_0$, choose stepsizes $\{\eta_{x_t}\}_{t=1}^T, \{\eta_{y_t}\}_{t=1}^T, \{\eta_{m_t}\}_{t=1}^T, \eta_z$ and $z_0$.
2: **for** $t = 1, \cdots, T$ **do**
3:      Initial $z_t^0 = z_0$, sample $\zeta_t^{(1)}, \cdots, \zeta_t^{(K)}, \xi_t^{(1)}$;
4:      **for** $k = 1, \cdots, K$ **do**
5:          $z_t^k = z_t^{k-1} - \eta_z(\nabla_{yy}^2 g(x_{t-1}, y_{t-1}, \zeta_t^{(k)})z_t^{k-1} - \nabla_y f(x_{t-1}, y_{t-1}, \xi_t^{(1)}))$;
6:      **end for**
7:      Sample $\zeta_t^{(K+1)}, \zeta_t^{(K+2)}$;
8:      $y_t = y_{t-1} - \eta_{y_t}(\nabla_y g(x_{t-1}, y_{t-1}, \zeta_t^{(K+1)}))$;
9:      $m_t = (1 - \eta_{m_t})m_{t-1} + \eta_{m_t}(\nabla_x f(x_{t-1}, y_{t-1}, \xi_t^{(1)}) - \nabla_{xy}^2 g(x_{t-1}, y_{t-1}, \zeta_t^{(K+2)})z_t^K)$
10:      $x_t = x_{t-1} - \eta_{x_t}m_t$
11: **end for**
12: Output $x_T, y_T$;

---

**Algorithm 2** ITD Bi-level Optimization Algorithm

---
1: Initilize $x_0$, choose stepsizes $\{\eta_{x_t}\}_{t=1}^T, \{\eta_{y_k}\}_{k=1}^K, y_0$.
2: **for** $t = 1, \cdots, T$ **do**
3:      Initial $y_t^0 = y_0$;
4:      **for** $k = 1, \cdots, K$ **do**
5:          Sample $\zeta_t^{(k)}$;
6:          $y_t^k = y_t^{k-1} - \eta_{y_k}(\nabla_y g(x_{t-1}, y_t^{k-1}, \zeta_t^{(k)}))$;
7:      **end for**
8:      Sample $\xi_t^{(1)}$
9:      $g_t = \nabla_x f(x_{t-1}, y_t^K, \xi_t^{(1)}) - \frac{\partial y_t^K}{\partial x_{t-1}}\nabla_y f(x_{t-1}, y_t^K, \xi_t^{(1)})$
10:      $x_t = x_{t-1} - \eta_{x_t}g_t$
11: **end for**
12: Output $x_T, y_T^K$;

---

**Proposition 1** (Lemma 2.1 in Ghadimi and Wang (2018)). *The gradient of the function $\Phi(x)$ can be given as*

$$\nabla\Phi(x) = \frac{1}{n}\sum_{i=1}^n \nabla_x f(x, y^*(x), \xi_i)$$

$$- \left(\frac{1}{q}\sum_{j=1}^q \nabla_{xy}^2 g(x, y^*(x), \zeta_j)\right)\left(\frac{1}{q}\sum_{j=1}^q \nabla_{yy}^2 g(x, y^*(x), \zeta_j)\right)^{-1}\left(\frac{1}{n}\sum_{i=1}^n \nabla_y f(x, y^*(x), \xi_i)\right).$$

This proposition is derived from the Implicit Function Theorem, a foundational concept in calculus. Consequently, we name the algorithm based on this proposition as the Approximate Implicit Differentiation (AID)-based method. The operation of this algorithm involves a sequence of updates, which are performed as follows:

Initially, we approximate $y^*(x_{t-1})$ with $y_{t-1}$, and we use $z_t^K$ to approximate $(\frac{1}{q}\sum_{j=1}^q \nabla_{yy}^2 g(x, y^*(x), \zeta_j))^{-1}(\frac{1}{n}\sum_{i=1}^n \nabla_y f(x, y^*(x), \xi_i))$. This approximation is formulated as a minimization problem with a quadratic objective function. We solve this quadratic function using Stochastic Gradient Descent (SGD) and then perform another round of SGD on $y$ and SGD with momentum on $x$. The AID algorithm is shown as the Algorithm 1.

Contrarily, the ITD-based methods adopt a different approach. These methods approximate the gradient of $x$ using the chain rules. Here, $y^*(x)$ is approximated by performing several gradient iterations. Therefore, in each iteration, we first update $y$ through several iterations of SGD from an initial point, followed by calculating the gradient of $x$ based on the chain rules. The ITD-based algorithm is shown as the Algorithm 2.

When observing Algorithm 2, the term $y_t^K$ can be expressed as a function of $x_{t-1}$, simplifying things significantly. This delightful peculiarity allows us to transform the analysis of ITD-based algorithms into the analysis of a simpler, single-level optimization problem. The only price we pay is a slight modification to the Lipschitz and smoothness constant.

In contrast, the landscape of Algorithm 1 is a little more intricate. The term $y_t$ can not be written directly in terms of $x_{t-1}$. Instead, it insists on drawing influence from the previous iteration of $x$. Likewise, $x_t$ doesn't simply depend on $y_{t-1}$, it keeps a record of all previous iterations, adding to the complexity. Moreover, the stability analysis of AID-based methods involves two other variable sequences $z_t^k$ and $m_t$. Both of them increase the difficulty of stability analysis.

## 3.2 GENERALIZATION DECOMPOSITION

In most cases involving bi-level optimization, there are two datasets: one in the upper-level problem and the other in the lower-level problem. The upper-level dataset is similar to the test data but has only a few data samples, and it's mainly used for validation. The lower-level dataset is usually a training dataset, and it may not have the same data distribution as the test data, but it contains a large number of samples. Because of the similarity and the number of samples of the upper-level dataset, our main focus is on achieving good generalization in the upper-level problem. Similar to the approach in Hardt et al. (2016), we define $\mathcal{A}(D_t, D_v)$ as the output of a bi-level optimization algorithm. For all training sets $D_t$, we can break down the generalization error as follows:

$$\mathbb{E}_{z, \mathcal{A}, D_v} f(\mathcal{A}(D_t, D_v), z) - \mathbb{E}_z f(x^*, y^*, z)$$

$$\leq \underbrace{\mathbb{E}_{z, \mathcal{A}, D_v} f(\mathcal{A}(D_t, D_v), z) - \mathbb{E}_{\mathcal{A}, D_v} \left[ \frac{1}{n} \sum_{i=1}^n f(\mathcal{A}(D_t, D_v), \xi_i) \right]}_{(I)}$$

$$+ \underbrace{\mathbb{E}_{\mathcal{A}, D_v} \left[ \frac{1}{n} \sum_{i=1}^n f(\mathcal{A}(D_t, D_v), \xi_i) \right] - \mathbb{E}_{D_v} \left[ \frac{1}{n} \sum_{i=1}^n f(\bar{x}, \bar{y}, \xi_i) \right]}_{(II)}$$

$$+ \underbrace{\mathbb{E}_{D_v} \left[ \frac{1}{n} \sum_{i=1}^n f(\bar{x}, \bar{y}, \xi_i) \right] - \mathbb{E}_{D_v} \left[ \frac{1}{n} \sum_{i=1}^n f(x^*, y^*, \xi_i) \right]}_{(III)} + \underbrace{\mathbb{E}_{D_v} \left[ \frac{1}{n} \sum_{i=1}^n f(x^*, y^*, \xi_i) \right] - \mathbb{E}_z f(x^*, y^*, z)}_{(IV)}$$

where $\bar{x}, \bar{y} \in \arg\min_{x, y^*(x)} \left\{ \frac{1}{n} \sum_{i=1}^n f(x, y^*(x), \xi_i), \ s.t. \ y^*(x) \in \arg\min_y \frac{1}{q} \sum_{j=1}^q g(x, y, \zeta_j) \right\}$, $x^*, y^* \in \arg\min_{x, y^*(x)} \left\{ \mathbb{E}_z f(x, y^*(x), z), \ s.t. \ y^*(x) \in \arg\min_y \frac{1}{q} \sum_{j=1}^q g(x, y, \zeta_j) \right\}$, $\xi_i$'s are the samples in the dataset $D_t$, and $\zeta_j$'s are the samples in the dataset $D_v$.

**Proposition 2** (Theorem 2.2 in Hardt et al. (2016))**.** *When for all $D_v$ and $D_v'$ which differ from 1 sample and for all $D_t$, $\sup_z f(\mathcal{A}(D_t, D_v), z) - f(\mathcal{A}(D_t, D_v'), z) \leq \epsilon$, we can obtain*

$$\mathbb{E}_{z, \mathcal{A}, D_v} f(\mathcal{A}(D_t, D_v), z) - \mathbb{E}_{\mathcal{A}, D_z} \left[ \frac{1}{n} \sum_{i=1}^n f(\mathcal{A}(D_t, D_v), \xi_i) \right] \leq \epsilon.$$

*Thus, with Proposition 2, we can bound term (I) by bounding $\sup_z f(\mathcal{A}(D_t, D_v), z) - f(\mathcal{A}(D_t, D_v'), z)$, as we'll explain in Section 4.2. Term (II) is an optimization error, and we'll control it in Section 4.3. Term (III) is less than or equal to 0 because of the optimality condition. Term (IV) is 0 when each sample in $D_v$ comes from the same distribution as $z$ independently.*

## 4 THEORETICAL ANALYSIS

In this section, we will give the theoretical results of Algorithm 1. Our investigation encompasses the stability and convergence characteristics of this algorithm and further explores the implications of various stepsize selections. We aim to ascertain the stability of Algorithm 1 when it attains an $\epsilon$-accuracy solution (i.e. $\mathbb{E}\|\nabla\Phi(x)\|^2 \leq \epsilon$, for some random vairable $x$).

## 4.1 BASIC ASSUMPTIONS AND DEFINITIONS

Our analysis begins with an examination of the stability of Algorithm 1. To facilitate this, we first establish the required assumptions for stability analysis.

**Assumption 1.** *Function $f(\cdot, \cdot, \xi)$ is lower bounded by $\underline{f}$ for all $\xi$. $f(\cdot, \cdot, \xi)$ is $L_0$-Lipschitz wih $L_1$-Lipschitz gradients for all $\xi$, i.e.*

$$|f(x_1, y, \xi) - f(x_2, y, \xi)| \leq L_0\|x_1 - x_2\|, \qquad |f(x, y_1, \xi) - f(x, y_2, \xi)| \leq L_0\|y_1 - y_2\|,$$
$$\|\nabla_x f(x_1, y, \xi) - \nabla_x f(x_2, y, \xi)\| \leq L_1\|x_1 - x_2\|, \quad \|\nabla_x f(x, y_1, \xi) - \nabla_x f(x, y_2, \xi)\| \leq L_1\|y_1 - y_2\|,$$
$$\|\nabla_y f(x_1, y, \xi) - \nabla_y f(x_2, y, \xi)\| \leq L_1\|x_1 - x_2\|, \quad \|\nabla_y f(x, y_1, \xi) - \nabla_y f(x, y_2, \xi)\| \leq L_1\|y_1 - y_2\|.$$

**Assumption 2.** *For all $x$ and $\zeta$, $g(x, \cdot, \zeta)$ is a $\mu$-strongly convex function with $L_1$-Lipschitz gradients:*

$$\|\nabla_y g(x, y_1, \zeta) - \nabla_y g(x, y_2, \zeta)\| \leq L_1 \|y_1 - y_2\|, \ \|\nabla_y g(x_1, y, \zeta) - \nabla_y g(x_2, y, \zeta)\| \leq L_1 \|x_1 - x_2\|.$$

*Further, for all $\zeta$, $g(\cdot, \cdot, \zeta)$ is twice-differentiable with $L_2$-Lipschitz second-order derivative i.e.,*

$$\|\nabla_{xy}^2 g(x_1, y, \zeta) - \nabla_{xy}^2 g(x_2, y, \zeta)\| \leq L_2 \|x_1 - x_2\|, \quad \|\nabla_{xy}^2 g(x, y_1, \zeta) - \nabla_{xy}^2 g(x, y_2, \zeta)\| \leq L_2 \|y_1 - y_2\|,$$
$$\|\nabla_{yy}^2 g(x_1, y, \zeta) - \nabla_{yy}^2 g(x_2, y, \zeta)\| \leq L_2 \|x_1 - x_2\|, \quad \|\nabla_{yy}^2 g(x, y_1, \zeta) - \nabla_{yy}^2 g(x, y_2, \zeta)\| \leq L_2 \|y_1 - y_2\|$$

These assumptions are in line with the standard requirements in the analysis of bi-level optimization (Ghadimi and Wang, 2018) and stability (Bao et al., 2021).

Subsequently, we define stability and elaborate its relationship with other forms of stability definitions.

**Definition 1.** *A bi-level algorithm $\mathcal{A}$ is $\beta$-stable iff for all $D_v, D_{v'} \in \mathcal{Z}_v^n$ such that $D_v, D_{v'}$ differ at most one sample, we have*

$$\forall D_t \in \mathcal{Z}_t^q, \mathbb{E}_{\mathcal{A}}[\|\mathcal{A}(D_t, D_v) - \mathcal{A}(D_t, D_{v'})\|] \leq \beta.$$

To compare with Bao et al. (2021), we first provide the stability definition in Bao et al. (2021).

**Definition 2** (Uniformly stability in Bao et al. (2021))**.** *A bi-level algorithm $\mathcal{A}$ is $\beta$-uniformly stable in expectation if the following inequality holds with $\beta \geq 0$:*

$$\left| \mathbb{E}_{\mathcal{A}, D_v \sim P_{D_v}^n, D_v' \sim P_{D_v}^n} \left[ f(\mathcal{A}(D_t, D_v), z) - f(\mathcal{A}(D_t, D_v'), z) \right] \right| \leq \beta, \ \forall D_t \in \mathcal{Z}_t^q, z \in Z_v.$$

The following proposition illustrates the relationship between our stability definition and the stability definition in Bao et al. (2021). They are only differentiated by a constant.

**Proposition 3.** *If algorithm $\mathcal{A}$ is $\beta$-stable, then it is $L_0\beta$-uniformly stable in expectation, where $L_0$ is Lipschitz constant for function f.*

**Remark 1.** *Consider the following simple hyperparameter optimization task where we employ ridge regression for the training phase. Let $x$ denote the regularization coefficient, $A_t$ the training input set, $A_v$ the validation input set, $b_t$ the training labels, $b_v$ the validation labels, and $y$ represent the model parameters. Thus, the bilevel optimization problem can be formulated as:*

$$\min_{x, y^*(x)} \left\{ \frac{1}{2} \|A_v y^*(x) - b_v\|^2, \ s.t. \ y^*(x) = \arg\min_y \frac{1}{2} \|A_t y - b_t\|^2 + \frac{x}{2} \|y\|^2. \right\}.$$

*The optimal solution for $y$ under a given $x$, denoted as $y^*(x)$, can be expressed as $y^*(x) = (A_t^T A_t + xI)^{-1} A_t^T b_t$. By substituting this solution into the upper-level optimization problem, we obtain:*

$$\min_x \frac{1}{2} \|A_v (A_t^T A_t + xI)^{-1} A_t^T b_t - b_v\|^2.$$

*This function is nonconvex with respect to $x$. Therefore, absent any additional terms in the upper-level optimization problem, the bilevel optimization problem is likely to have a nonconvex objective with respect to $x$. As such, we make no assumptions about convexity in relation to $x$. Importantly, we refrain from introducing additional terms to the upper-level problem as it could lead to the inclusion of new hyperparameters that need to be further tunned.*

## 4.2 STABILITY OF ALGORITHM 1

In this part, we present our stability findings for the AID-based bilevel optimization algorithm 1.

**Theorem 1.** *Suppose assumptions 1 and 2 hold, Algorithm 1 is $\epsilon_{stab}$-stable, where*

$$\epsilon_{stab} = \sum_{t=1}^T \Pi_{k=t+1}^T (1 + \eta_{x_k} \eta_{m_k} C_m + \eta_{m_k} C_m + \eta_{y_k} L_1)(1 + \eta_{x_t}) \eta_{m_t} C_c / n,$$

$$C_m = \frac{2(n-1)L_1}{n} + 2L_2 D_z + \frac{L_1}{\mu} \left( \frac{(n-1)L_1}{n} + D_z L_2 \right)$$

$$D_z = (1 - \mu\eta_z)^K \|z_0\| + \frac{L_0}{\mu}, C_c = 2L_0 + \frac{2L_1 L_0}{\mu}.$$

**Corollary 1.** *Suppose assumption 1, 2 hold and that $f(x, y, \xi) \in [0, 1]$, by selecting $\eta_{x_t} = \eta_{m_t} = \alpha/t$, $\eta_{y_t} = \beta/t$, Algorithm 1 is $\epsilon_{stab}$-stable, where*

$$\epsilon_{stab} = \mathcal{O}\left(T^q/n\right),$$

$q = \frac{2C_m\alpha + L_1\beta}{2C_m\alpha + L_1\beta + 1} < 1$, $C_m = \frac{2(n-1)L_1}{n} + 2L_2D_z + \frac{L_1}{\mu}\left(\frac{(n-1)L_1}{n} + D_zL_2\right)$ and $D_z = (1 - \mu\eta_z)^K\|z_0\| + \frac{L_0}{\mu}$.

**Remark 2.** *The results in Bao et al. (2021), show ITD-based methods achieve $\mathcal{O}\left(\frac{T^\kappa}{n}\right)$, for some $\kappa < 1$. Moreover, Hardt et al. (2016) show the uniform stability in nonconvex single-level optimization with the order of $\mathcal{O}\left(\frac{T^k}{n}\right)$ where $k$ is a constant less than 1. We achieve the same order of sample size and similar order on the number of iterations.*

### 4.3 CONVERGENCE ANALYSIS

To give an analysis of convergence, we further give the following assumption.

**Assumption 3.** *For all $x, y$, there exists $D_0, D_1$ such that the following inequality holds:*

$$\frac{1}{q}\sum_{j=1}^{q}\left\|\nabla_y g(x, y, \xi_j) - \left(\frac{1}{q}\sum_{j=1}^{q}\nabla_y g(x, y, \xi_j)\right)\right\|^2 \leq D_1\left\|\frac{1}{q}\sum_{j=1}^{q}\nabla_y g(x, y, \xi_j)\right\|^2 + D_0$$

This assumption is a generalized assumption of bounded variance in stochastic gradient descent. When $D_1 = 1$, $D_0$ can be viewed as the variance of the stochastic gradient. When $D_0 = 0$, and $D_1 > 1$, it is called strong growth condition, which shows the ability of a large-scale model that can represent each data well.

Given specific conditions of $\eta_{m_t}$, $\eta_{x_t}$ and $\eta_{y_t}$, we present the following convergence results.

**Theorem 2.** *Suppose the Assumptions 1, 2 and 3 hold, and the following conditions are satisfied:*

$$\frac{\eta_{x_t}}{\eta_{y_t}} \leq \frac{\mu}{4L_1(L_1 + D_2L_2)}, \eta_{x_t} \leq \frac{1}{2L_\Phi}, \eta_z \leq \frac{1}{L_1} \tag{2}$$

*and $\eta_{m_t}$, $\frac{\eta_{m_t}}{\eta_{x_t}}$ and $\frac{\eta_{m_t}}{\eta_{y_t}}$ are non-increasing, where $L_\Phi = \frac{(\mu+L_1)\left(L_1\mu^2 + L_0L_2\mu + L_1^2\mu + L_2L_0\right)}{\mu^3}$. Define $\Phi(x) = \frac{1}{n}\sum_{i=1}^{n} f(x, y^*(x), \xi_i)$, where $y^*(x) = \arg\min_y \frac{1}{q}\sum_{j=1}^{q} g(x, y, \zeta_j)$. Then, when $K = \Theta(\log T)$, it holds that*

$$\min_{t\in\{1,\cdots,T\}} \mathbb{E}\|\nabla\Phi(x_t)\|^2 = \mathcal{O}\left(\frac{1 + \sum_{k=1}^{T}\eta_{y_k}\eta_{m_k} + \eta_{m_k}^2}{\sum_{k=1}^{T}\eta_{m_k}}\right).$$

**Remark 3.** *When we set $\eta_{x_t} = \Theta(1/\sqrt{T})$, $\eta_{m_t} = \Theta(1/\sqrt{T})$ and $\eta_{y_t} = \Theta(1/\sqrt{T})$, we achieve a convergence rate of $\mathcal{O}(1/\sqrt{T})$, which aligns with the bound of the SGD momentum algorithm in single-level optimization problems. Thus, the convergence upper bound seems plausible.*

### 4.4 TRADE-OFF IN GENERALIZATION ABILITY

After determining the convergence of Algorithm 1 and its stability, we can derive the following corollary using the learning rate typically employed in non-convex stability analysis.

**Corollary 2.** *When we choose $\eta_{x_t} = \Theta(1/t)$, $\eta_{m_t} = \Theta(1/t)$, and $\eta_{y_t} = \Theta(1/t)$, by satisfying the conditions in Theorem 2, it holds that when $\min_{t\in\{1,\cdots,T\}} \mathbb{E}\|\nabla\Phi(x_t)\|^2 \leq \epsilon$, $\log \epsilon_{stab} = \mathcal{O}(1/\epsilon)$.*

**Remark 4.** *Although we can get a good stability bound when using the learning rate with the order $1/t$, it suffers from its convergence rate, which is $\mathcal{O}(1/\log T)$. Thus, with the learning rate in the order of $1/t$, we can only get stability at an exponential rate to achieve some $\epsilon$-accuaracy solution.*

In practice, a constant learning rate is often used for $T$ iterations, leading to the following corollary.

**Corollary 3.** *When we choose $\eta_{x_t} = \eta_x, \eta_{m_t} = \eta_m, \eta_{y_t} = \eta_y$ for some postive constant $\eta_x, \eta_m$ and $\eta_y$. Then it holds that when $\min_{t\in\{1,\cdots,T\}} \mathbb{E}\|\nabla\Phi(x_t)\|^2 \leq \epsilon$, the upper bound of $\log \epsilon_{stab}$ is at least in the order of $1/\epsilon$.*

**Remark 5.** *Although with some constant stepsize related to $T$, the convergence rate could be much faster than $\mathcal{O}(1/\log T)$, the stability will explode up quickly, which leads the increase of stability at an exponential rate.*

**Remark 6.** *From the above two corollaries, in practice, a diminishing learning rate is often preferable due to its stronger theoretical generalization ability.*

### 4.5 PROOF SKETCH

In this subsection, we illustrate the proof sketches for our main theorems and corollaries. Furthermore, several useful lemmas are also introduced.

#### 4.5.1 PROOF SKETCH FOR THEOREM 1

To prove Theorem 1, we first define some notations and give several lemmas.

**Notation 1.** *We use $x_t, y_t, z_t^k$ and $m_t$ to represent the iterates in Algorithm 1 with dataset $D_v$ and $D_t$. We use $\tilde{x}_t, \tilde{y}_t, \tilde{z}_t^k$ and $\tilde{m}_t$ to represent the iterates in Algorithm 1 with dataset $D_v'$ and $D_t$.*

Then, we bound $\|x_t - \tilde{x}_t\|$, $\|y_t - \tilde{y}_t\|$, $\|m_t - \tilde{m}_t\|$ and $\|z_t - \tilde{z}_t\|$ by the difference of previous iteration (i.e. $\|x_{t-1} - \tilde{x}_{t-1}\|$, $\|y_{t-1} - \tilde{y}_{t-1}\|$, $\|m_{t-1} - \tilde{m}_{t-1}\|$) as the following 4 lemmas.

**Lemma 1.** *With the update rules defined in Algorithm 1, it holds that*

$$\mathbb{E}\|z_t^K - \tilde{z}_t^K\| \leq \mathbb{E}\left[\frac{1}{\mu}\left(\frac{(n-1)L_1}{n} + D_z L_2\right)(\|x_{t-1} - \tilde{x}_{t-1}\| + \|y_{t-1} - \tilde{y}_{t-1}\|)\right] + \frac{2L_0}{n\mu}.$$

**Lemma 2.** *With the update rules defined in Algorithm 1, it holds that*

$$\mathbb{E}\|y_t - \tilde{y}_t\| \leq \eta_{y_t} L_1 \mathbb{E}\|x_{t-1} - \tilde{x}_{t-1}\| + (1 - \mu\eta_{y_t}/2)\mathbb{E}\|y_{t-1} - \tilde{y}_{t-1}\|.$$

**Lemma 3.** *With the update rules defined in Algorithm 1, it holds that*

$$\mathbb{E}\|m_t - \tilde{m}_t\|$$

$$\leq \mathbb{E}\left[(1 - \eta_{m_t})\|m_{t-1} - \tilde{m}_{t-1}\| + \eta_{m_t} C_m(\|x_{t-1} - \tilde{x}_{t-1}\| + \|y_{t-1} - \tilde{y}_{t-1}\|)\right] + \eta_{m_t}\left(\frac{2L_0 + 2L_1 L_0}{n}\right),$$

*where $C_m = \frac{2(n-1)L_1}{n} + 2L_2 D_z + \frac{L_1}{\mu}\left(\frac{(n-1)L_1}{n} + D_z L_2\right)$.*

**Lemma 4.** *With the update rules defined in Algorithm 1, it holds that*

$$\mathbb{E}\|x_t - \tilde{x}_t\| \leq \mathbb{E}\left[(1 + \eta_{x_t}\eta_{m_t} C_m)\|x_{t-1} - \tilde{x}_{t-1}\| + \eta_{x_t}\eta_{m_t} C_m\|y_{t-1} - \tilde{y}_{t-1}\|\right]$$

$$+ \mathbb{E}\left[\eta_{x_t}(1 - \eta_{m_t})\|m_{t-1} - \tilde{m}_{t-1}\|\right] + \eta_{x_t}\eta_{m_t}\left(\frac{2L_0 + 2L_1 L_0}{n}\right),$$

*where $C_m = \frac{2(n-1)L_1}{n} + 2L_2 D_z + \frac{L_1}{\mu}\left(\frac{(n-1)L_1}{n} + D_z L_2\right)$.*

The last step was to combine the above 4 lemmas, by induction and some calculation, then we can obtain the result in Theorem 1.

#### 4.5.2 PROOF SKETCH FOR THEOREM 2

In fact, Chen et al. (2022) recently have given the convergence results for AID-based bilevel optimization with constant learning rate $\eta_x$, $\eta_m$, and $\eta_y$. Theorem 2 can be regarded as an extended version of that in Chen et al. (2022) with time-evolving learning rates. To show the proofs, we first give the descent lemma for $x$ and $y$ with the general time-evolving learning rates.

**Lemma 5.** *With the update rules of $y_t$ it holds that*

$$\mathbb{E}\|y_t - y^*(x_t)\|^2 \leq (1 - \mu\eta_{y_t}/2)\mathbb{E}\|y_{t-1} - y^*(x_{t-1})\|^2 + \frac{(2 + \mu\eta_{y_t})L_1^2\eta_{x_t}^2}{\mu\eta_{y_t}}\mathbb{E}\|m_t\|^2 + 2\eta_{y_t}^2 D_0$$

**Lemma 6.** *With the update rules of $x_t$ and $m_t$, it holds that*

$$\mathbb{E}\left[\frac{\eta_{m_t}}{\eta_{x_t}}\Phi(x_t) + \frac{1 - \eta_{m_t}}{2}\|m_t\|^2 - \frac{\eta_{m_t}}{\eta_{x_t}}\Phi(x_{t-1}) - \frac{1 - \eta_{m_t}}{2}\|m_{t-1}\|^2\right]$$

$$\leq \eta_{m_t}(L_1 + D_z L_2)^2\mathbb{E}\|y_{t-1} - y^*(x_{t-1})\|^2 + \eta_{m_t}L_1^2(1 - \eta_x\mu)^{2K}\left(D_z + \frac{L_0}{\mu}\right)^2 - \frac{\eta_{m_t}}{4}\mathbb{E}\|m_t\|^2.$$

Then, combining two descent lemmas, we can show that $\liminf_{t\to\infty} \mathbb{E}\|m_t\|^2 = 0$. The last step is to establish the relation between $m_t$ and $\nabla\Phi(x_t)$, which is given by the following lemma.

**Lemma 7.** *With the update rules of $m_t$, it holds that*

$$\sum_{t=1}^{T} \eta_{m_{t+1}} \mathbb{E}\|m_t - \nabla\Phi(x_t)\|^2 \le \mathbb{E}\|\nabla\Phi(x_0)\|^2 + \sum_{t=1}^{T} 2\eta_{m_t}\mathbb{E}\|\mathbb{E}\Delta_t - \nabla\Phi(x_{t-1})\|^2$$
$$+ 2\eta_{x_t}^2/\eta_{m_t} L_1^2 \|m_t\|^2 + \eta_{m_t}^2 \mathbb{E}\|\Delta_t - \mathbb{E}\Delta_t\|^2.$$

*where* $\Delta_t = \nabla_x f(x_{t-1}, y_{t-1}, \xi_t^{(1)}) - \nabla_{xy}^2 g(x_{t-1}, y_{t-1}, \zeta_t^{(K+2)}) z_t^K$.

As the variance can be shown bounded, the error for gradient estimation can be small when K is large. we can give the convergence of Algorithm 1 under the conditions in Theorem 2.

## 5 EXPERIMENTS

In this section, we conduct two kinds of experiments to verify our theoretical findings.

### 5.1 TOY EXAMPLE

To illustrate the practical application of our theoretical framework, we tackle a simplified case of transfer learning, where the source domain differs from the target domain by an unknown linear transformation X. The problem is formulated as follows:

$$\min_{X} \quad \frac{1}{n}\sum_{i=1}^{n} \|A_2(i)y^*(X) - b_2(i)\|^2 + \rho_1\|X^TX - I\|^2$$

$$s.t.\ y^*(X) \in \arg\min \frac{1}{q}\sum_{j=1}^{q} \|A_1(j)Xy - b_1(j)\|^2 + \rho_2\|y\|^2,$$

Here, $A_2(i)$ and $A_1(j)$ represent the $i$-th row and $j$-th row of matrices $A_2$ and $A_1$, respectively. $A_1 \in \mathbb{R}^{2000\times10}$, $A_2 \in \mathbb{R}^{n\times10}$ are randomly generated from a Gaussian distribution with mean 0 and variance 0.05. Employing a ground truth unitary matrix $\hat{X}^{10\times10}$ and a vector $\hat{y} \in \mathbb{R}^{10}$, we generate $b_1 = A_1\hat{X}\hat{y} + n_1$, $b_2 = A_2\hat{y} + n_2$, where $n_1, n_2$ are independent Gaussian noise with variance 0.1. We test for $n$ in the set $\{500, 1000\}$. For constant learning rates, we select it from $\{0.01, 0.005, 0.001\}$, while for diminishing learning rates, we select a constant from $\{1000, 2000\}$ and the learning rate from $\{1, 2, 5, 10\}$, and set the learning rate as $initial\_learning\_rate/(iteration + constant)$. We fix $K = 10$ and $\eta_z = 0.01$ for all experiments.

To evaluate the results, we employ the function value of the upper-level objective as the optimization error, and the difference between the output $X$ and ground truth $\hat{X}$ as the generalization error. Each experiment is run for five times, with the averaged results shown in Figure 1.

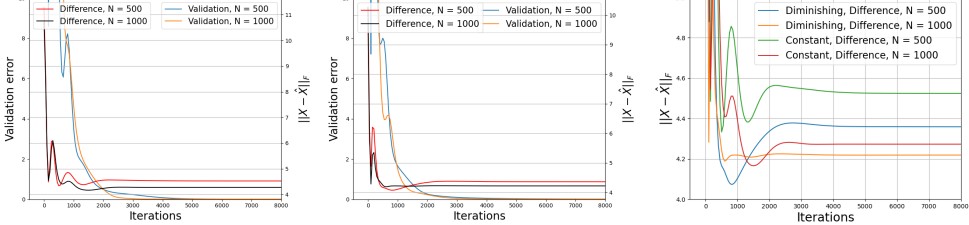

Figure 1: Results for Toy Example. The left figure shows the results when learning rates are constant, the middle figure shows the results when we use diminishing learning rates, and the right figure compares the results for constant learning rates and diminishing learning rates.

Upon examining the results, it becomes apparent that even when the function value of the upper-level objective approaches zero, a noticeable discrepancy exists between the output $X$ and the ground truth $\hat{X}$. However, encouragingly, as we increase the number of points ($n$) in the validation set, this gap begins to shrink. This is a finding that is in line with the predictions made in Theorem 1. A closer comparison between the algorithm employing a constant learning rate and the one with a diminishing learning rate reveals another significant observation. The diminishing learning rate approach yields

smaller gaps, thus enhancing generalization performance. This experimental outcome substantiates the assertions made in Corollary 2 and Corollary 3, demonstrating that the generalization ability for diminishing learning rates outperforms the generalization ability for constant rates when aiming to achieve a certain optimization accuracy.

## 5.2 DATA SELECTION ON MNIST

We apply Algorithm 1 on MNIST (Deng, 2012), a resource composed of 60,000 digit recognition samples at a resolution of $28 \times 28$. The task is to identify and select valuable data within the dataset.

We structure our experiment as follows. We designate $n$ data samples from the training dataset to serve as a validation dataset. Concurrently, we randomly select 5,000 samples from the remaining training set to establish a new training set, with half of these samples randomly labeled. For classification, we employ LeNet5 (LeCun et al., 1998) model as the backbone. Our experiment is based on a bi-level optimization problem, defined as follows:

$$\min_{x,y^*(x)} \frac{1}{n} \sum_{i=1}^{n} L(f(y^*(x), \xi_{i,input}), \xi_{i,label})$$

$$s.t. y^*(x) \in \arg\min_{y} \frac{1}{q} \sum_{j=1}^{q} x_j L(f(y, \zeta_{j,input}), \zeta_{j,label}), \ 0 \le x_j \le 1$$

Here, $f$ represents the LeNet5 model, $L$ denotes the cross-entropy loss, $\xi_i$ is a sample from the validation set, and $\zeta_j$ is a sample from the new training set. We put our algorithm to the test under both diminishing and constant learning rates, using varying validation sizes of $n \in \{100, 200\}$. Learning rates for the constant learning rate are selected from the set $\{0.1, 0.05, 0.001\}$, while for the diminishing learning rate, the constants are chosen from $\{200, 300, 400\}$ and learning rates from $\{5, 10, 20, 30, 40\}$, where the learning rate of each component is calculated by $initial\_learning\_rates/(iterations + constant)$. All experiments maintain $K = 2$ and $\eta_z = 0.1$. Each experiment is run for five times, with averaged results shown in Figure 2.

As can be observed from the figure, even with a 100% accuracy rate on the validation set, a noticeable gap persists between test accuracy and validation accuracy. As we incrementally increase the number of samples in the validation set, we notice an encouraging trend: the accuracy of the test set improves for both constant and diminishing learning rates. This finding aligns with our predictions in Theorem 1. Moreover, the implementation of a diminishing learning rate yields a higher test accuracy, indicating a smaller generalization gap. This observation aligns with our theoretical findings as outlined in Corollary 2 and Corollary 3, thus validating our theoretical assertions with empirical evidence.

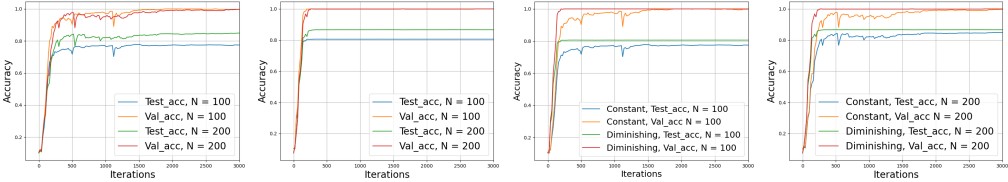

Figure 2: Results for Data selection on MNIST. The first figure shows the result with constant learning rates. The second figure shows the results with diminishing learning rates. The third figure and fourth figure compare the results between constant learning rates and diminishing learning rates with 100 samples in the validation set and 200 samples in the validation set, respectively.

## 6 CONCLUSION

In this paper, we have ventured into the realm of stability analysis, specifically focusing on an AID-based bi-level algorithm. Our findings have produced results of comparable order to those derived from ITD-based methods and single-level non-convex SGD techniques. Our exploration extended to convergence analysis under specific conditions for stepsize selection. An intriguing interplay between convergence analysis and stability was revealed, painting a compelling theoretical picture that favors diminishing stepsize over its constant counterpart. The empirical evidence corroborates our theoretical deductions, providing tangible validation for our assertions. However, there is still a mystery for the proper choice of stepsize, and for the weaker conditions of the upper-level and lower-level objective function, we will leave for future work.

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

# A    PROOF OF THEOREM 1

**Notation 2.** *We use $x_t, y_t, z_t^k$ and $m_t$ to represent the iterates in Algorithm 1 with dataset $D_v$ and $D_t$. We use $\tilde{x}_t, \tilde{y}_t, \tilde{z}_t^k$ and $\tilde{m}_t$ to represent the iterates in Algorithm 1 with dataset $D_v'$ and $D_t$.*

**Lemma 8.** *With the assumption 1 and 2, by selecting $\eta_z \leq 1/L_1$, it holds that $\left\|z_t^k\right\| \leq D_z$ for all $k$, where $D_z = \|z_0\| + \frac{L_0}{\mu}$ .*

*Proof.* Let $b = \nabla_y f\left(x_{t-1}, y_{t-1}, \xi_t^{(1)}\right)$.

With Assumption 1 and 2, it holds that

$$\mu I \preceq A \preceq L_1 I, \text{ and }, \|b\| \leq L_0.$$

According to Algorithm 1, it holds that

$$z_t^k = z_t^{k-1} - \eta_z \left(\nabla_{yy}^2 g\left(x_{t-1}, y_{t-1}, \zeta_t^{(k)}\right) z_t^{k-1} - b\right) = \left(I - \eta_z \nabla_{yy}^2 g\left(x_{t-1}, y_{t-1}, \zeta_t^{(k)}\right)\right) z_t^{k-1} + \eta_z b.$$

Thus, it holds that

$$\begin{aligned}
\left\|z_t^k\right\| &\leq \left\|I - \eta_z \nabla_{yy}^2 g\left(x_{t-1}, y_{t-1}, \zeta_t^{(k)}\right)\right\| \left\|z_t^{k-1}\right\| + L_0 \eta_z \\
&\leq (1 - \mu\eta_z) \left\|z_t^{k-1}\right\| + L_0 \eta_z \\
&\leq \cdots \leq (1 - \mu\eta_z)^k \|z_0\| + \sum_{t=0}^{k-1} (1 - \mu\eta_z)^t L_0 \eta_z \\
&\leq (1 - \mu\eta_z)^k \|z_0\| + \frac{L_0}{\mu} \\
&\leq \|z_0\| + \frac{L_0}{\mu}.
\end{aligned}$$

Hence, we obtain the desired results. $\qquad\square$

**Lemma 9.** *With the update rules defined in Algorithm 1, it holds that*

$$\mathbb{E}\|z_t^K - \tilde{z}_t^K\| \leq \mathbb{E}\left[\frac{1}{\mu}\left(\frac{(n-1)L_1}{n} + D_z L_2\right)(\|x_{t-1} - \tilde{x}_{t-1}\| + \|y_{t-1} - \tilde{y}_{t-1}\|)\right] + \frac{2L_0}{n\mu}.$$

*Proof.* According to Algorithm 1, it holds that

$$\begin{aligned}
&\mathbb{E}\left\|z_t^k - \tilde{z}_t^k\right\| \\
&= \mathbb{E}\left\|z_t^{k-1} - \eta_z \left(\nabla_{yy}^2 g\left(x_{t-1}, y_{t-1}, \zeta_t^{(k)}\right) z_t^{k-1} - \nabla_y f\left(x_{t-1}, y_{t-1}, \xi_t^{(1)}\right)\right)\right. \\
&\quad \left. - \tilde{z}_t^{k-1} + \eta_z \left(\nabla_{yy}^2 \nabla g\left(\tilde{x}_{t-1}, \tilde{y}_{t-1}, \zeta_t^{(k)}\right) \tilde{z}_t^{k-1} - \nabla_y f\left(\tilde{x}_{t-1}, \tilde{y}_{t-1}, \tilde{\xi}_t^{(1)}\right)\right)\right\| \\
&\leq \mathbb{E}\left[\left\|\left(I - \eta_z \nabla_{yy}^2 g\left(x_{t-1}, y_{t-1}, \zeta_t^{(k)}\right)\right) z_t^{k-1} - \left(I - \eta_z \nabla_{yy}^2 g\left(\tilde{x}_{t-1}, \tilde{y}_{t-1}, \zeta_t^{(k)}\right)\right) \tilde{z}_t^{k-1}\right\| \right. \\
&\quad \left. + \eta_z \left\|\nabla_y f\left(x_{t-1}, y_{t-1}, \xi_t^{(1)}\right) - \nabla_y f\left(\tilde{x}_{t-1}, \tilde{y}_{t-1}, \tilde{\xi}_t^{(1)}\right)\right\|\right] \\
&\leq \mathbb{E}\left[\eta_z \left\|z_t^{k-1}\right\| \left\|\nabla_{yy}^2 g\left(x_{t-1}, y_{t-1}, \zeta_t^{(k)}\right) - \nabla_{yy}^2 g\left(\tilde{x}_{t-1}, \tilde{y}_{t-1}, \zeta_t^{(k)}\right)\right\| \right. \\
&\quad \left. + \left\|I - \eta_z \nabla_{yy}^2 g\left(\tilde{x}_{t-1}, \tilde{y}_{t-1}, \zeta_t^{(1)}\right)\right\| \left\|z_t^{k-1} - \tilde{z}_t^{k-1}\right\| + \eta_z \left\|\nabla_y f\left(x_{t-1}, y_{t-1}, \xi_t^{(1)}\right) - \nabla_y f\left(\tilde{x}_{t-1}, \tilde{y}_{t-1}, \tilde{\xi}_t^{(1)}\right)\right\|\right]
\end{aligned}$$

According to Lemma 8, we have $\|z_t^{k-1}\| \leq D_z$.

According to Assumption 1 and 2, we have the following inequalities:

$$\begin{aligned}
\left\|\nabla_{yy}^2 g\left(x_{t-1}, y_{t-1}, \zeta_t^{(1)}\right) - \nabla_{yy}^2 g\left(\tilde{x}_{t-1}, \tilde{y}_{t-1}, \zeta_t^{(1)}\right)\right\| &\leq L_2 (\|x_t - \tilde{x}_t\| + \|y_t - \tilde{y}_t\|) \\
\left\|I - \eta_z \nabla_{yy}^2 g\left(\tilde{x}_{t-1}, \tilde{y}_{t-1}, \zeta_t^{(1)}\right)\right\| &\leq (1 - \mu\eta_z)
\end{aligned}$$

For $\left\| \nabla_y f \left( x_{t-1}, y_{t-1}, \xi_t^{(1)} \right) - \nabla_y f \left( \tilde{x}_{t-1}, \tilde{y}_{t-1}, \tilde{\xi}_t^{(1)} \right) \right\|$, when $\tilde{\xi}_t^{(1)} \neq \xi_t^{(1)}$, which happens with probability $\frac{1}{n}$, it holds that $\left\| \nabla_y f \left( x_{t-1}, y_{t-1}, \xi_t^{(1)} \right) - \nabla_y f \left( \tilde{x}_{t-1}, \tilde{y}_{t-1}, \tilde{\xi}_t^{(1)} \right) \right\| \leq 2L_0$.

When $\tilde{\xi}_t^{(1)} \neq \xi_t^{(1)}$, which happens with probability $1 - \frac{1}{n}$, it holds that $\left\| \nabla_y f \left( x_{t-1}, y_{t-1}, \xi_t^{(1)} \right) - \nabla_y f \left( \tilde{x}_{t-1}, \tilde{y}_{t-1}, \tilde{\xi}_t^{(1)} \right) \right\| \leq L_1 \left( \| x_{t-1} - \tilde{x}_{t-1} \| + \| y_t - \tilde{y}_{t-1} \| \right)$.

Thus, combining the above inequalities, it holds that

$$\mathbb{E} \left\| z_t^k - \tilde{z}_t^k \right\|$$

$$\leq \mathbb{E} \left[ \eta_z \left\| z_t^{k-1} \right\| \left\| \nabla_{yy}^2 g \left( x_{t-1}, y_{t-1}, \zeta_t^{(1)} \right) - \nabla_{yy}^2 g \left( \tilde{x}_{t-1}, \tilde{y}_{t-1}, \zeta_t^{(1)} \right) \right\| \right.$$

$$\left. + \left\| I - \eta_z \nabla_{yy}^2 g \left( \tilde{x}_{t-1}, \tilde{y}_{t-1}, \zeta_t^{(1)} \right) \right\| \left\| z_t^{k-1} - \tilde{z}_t^{k-1} \right\| + \eta_z \left\| \nabla_y f \left( x_{t-1}, y_{t-1}, \xi_t^{(1)} \right) - \nabla_y f \left( \tilde{x}_{t-1}, \tilde{y}_{t-1}, \tilde{\xi}_t^{(1)} \right) \right\| \right]$$

$$\leq \mathbb{E} \left[ \eta_z D_z L_2 \left( \| x_{t-1} - \tilde{x}_{t-1} \| + \| y_{t-1} - \tilde{y}_{t-1} \| \right) + (1 - \eta_z \mu) \left\| z_t^{k-1} + \tilde{z}_t^{k-1} \right\| \right.$$

$$\left. + \frac{2\eta_z L_0}{n} + \left( 1 - \frac{1}{n} \right) \eta_z L_1 \left( \| x_{t-1} - \tilde{x}_{t-1} \| + \| y_{t-1} - \tilde{y}_{t-1} \| \right) \right]$$

$$= \mathbb{E} \left[ \eta_z \left( \frac{(n-1)L_1}{n} + D_z L_2 \right) \left( \| x_{t-1} - \tilde{x}_{t-1} \| + \| y_{t-1} - \tilde{y}_{t-1} \| \right) + (1 - \eta_z \mu) \left\| z_t^{k-1} - \tilde{z}_t^{k-1} \right\| \right] + \frac{2\eta_z L_0}{n}$$

$$\leq \cdots \leq \sum_{t=0}^{k} (1 - \eta_z \mu)^t \left[ \mathbb{E} \left[ \eta_z \left( \frac{(n-1)L_1}{n} + D_z L_2 \right) \left( \| x_{t-1} - \tilde{x}_{t-1} \| + \| y_{t-1} - \tilde{y}_{t-1} \| \right) \right] + \frac{2\eta_z L_0}{n} \right]$$

$$\leq \mathbb{E} \left[ \frac{1}{\mu} \left( \frac{(n-1)L_1}{n} + D_z L_2 \right) \left( \| x_{t-1} - \tilde{x}_{t-1} \| + \| y_{t-1} - \tilde{y}_{t-1} \| \right) \right] + \frac{2L_0}{n\mu}.$$

Hence, we get the desired result.

$\square$

**Lemma 10.** *With the update rules defined in Algorithm 1, it holds that*

$$\mathbb{E} \| y_t - \tilde{y}_t \| \leq \eta_{y_t} L_1 \| x_{t-1} - \tilde{x}_{t-1} \| + (1 - \mu \eta_{y_t}/2) \| y_{t-1} - \tilde{y}_{t-1} \|$$

*Proof.* With the update rules, it holds that

$$\mathbb{E} \| y_t - \tilde{y}_t \|$$

$$= \mathbb{E} \left\| y_{t-1} - \eta_{y_t} \nabla_y g \left( x_{t-1}, y_{t-1}, \zeta_t^{(K+1)} \right) - \tilde{y}_{t-1} + \eta_{y_t} \nabla_y g \left( \tilde{x}_{t-1}, \tilde{y}_{t-1}, \zeta_t^{(K+1)} \right) \right\|$$

$$\leq \mathbb{E} \left[ \left\| y_{t-1} - \eta_{y_t} \nabla_y g \left( x_{t-1}, y_{t-1}, \zeta_t^{(K+1)} \right) - \tilde{y}_{t-1} + \eta_{y_t} \nabla_y g \left( x_{t-1}, \tilde{y}_{t-1}, \zeta_t^{(K+1)} \right) \right\| \right.$$

$$\left. + \eta_{y_t} \left\| \nabla_y g \left( x_{t-1}, \tilde{y}_{t-1}, \zeta_t^{(K+1)} \right) - \nabla_y g \left( \tilde{x}_{t-1}, \tilde{y}_{t-1}, \zeta_t^{(K+1)} \right) \right\| \right]$$

With the strongly convexity of function $g(x, \cdot, \zeta)$, it holds that

$$\langle \nabla_y g(x, y_1, \zeta) - \nabla_y g(x, y_2, \zeta), y_1 - y_2 \rangle \geq \mu \| y_1 - y_2 \|^2.$$

Thus, it holds that

$$\left\| y_{t-1} - \eta_{y_t} \nabla_y g \left( x_{t-1}, y_{t-1}, \zeta_t^{(K+1)} \right) - \tilde{y}_{t-1} + \eta_{y_t} \nabla_y g \left( x_{t-1}, \tilde{y}_{t-1}, \zeta_t^{(K+1)} \right) \right\|^2$$

$$= \| y_{t-1} - \tilde{y}_{t-1} \|^2 - 2\eta_{y_t} \langle y_{t-1} - \tilde{y}_{t-1}, \nabla_y g \left( x_{t-1}, y_{t-1}, \zeta_t^{(K+1)} \right), -\nabla_y g \left( x_{t-1}, \tilde{y}_{t-1}, \zeta_t^{(2)} \right) \rangle$$

$$+ \eta_{y_t}^2 \| \nabla_y g \left( x_{t-1}, y_{t-1}, \zeta_t^{(2)} \right) - \nabla_y g \left( x_{t-1}, \tilde{y}_{t-1}, \zeta_t^{(2)} \right) \|^2$$

$$\leq \| y_{t-1} - \tilde{y}_{t-1} \|^2 - 2\mu \eta_{y_t} \| y_{t-1} - \tilde{y}_{t-1} \|^2 + L^2 \eta_{y_t}^2 \| y_{t-1} - \tilde{y}_{t-1} \|^2.$$

By selecting $\eta_y$ such that $\mu \eta_{y_t} \geq L^2 \eta_{y_t}^2$, we can obtain

$$\left\| y_{t-1} - \eta_{y_t} \nabla_y g \left( x_{t-1}, y_{t-1}, \zeta_t^{(K+1)} \right) - \tilde{y}_{t-1} + \eta_{y_t} \nabla_y g \left( x_{t-1}, \tilde{y}_{t-1}, \zeta_t^{(K+1)} \right) \right\|^2 \leq (1 - \mu \eta_y) \| y_{t-1} - \tilde{y}_{t-1} \|^2$$

With $\sqrt{1-x} \le 1 - x/2$ when $x \in [0,1]$, it holds that

$$\left\| y_{t-1} - \eta_{y_t}\nabla_y g\left(x_{t-1}, y_{t-1}, \zeta_t^{(K+1)}\right) - \tilde{y}_{t-1} + \eta_{y_t}\nabla_y g\left(x_{t-1}, \tilde{y}_{t-1}, \zeta_t^{(K+1)}\right)\right\| \le (1 - \mu\eta_y/2)\|y_{t-1} - \tilde{y}_{t-1}\|$$

On the other hand, with Assumption 2, it holds that

$$\left\|\nabla_y g\left(x_{t-1}, \tilde{y}_{t-1}, \zeta_t^{(K+1)}\right) - \nabla_y g\left(\tilde{x}_{t-1}, \tilde{y}_{t-1}, \zeta_t^{(K+1)}\right)\right\| \le L_1\|x_{t-1} - \tilde{x}_{t-1}\|$$

Thus, by combining the above inequalities, we can obtain that

$$\mathbb{E}\|y_t - \tilde{y}_t\| \le \eta_{y_t}L_1\|x_{t-1} - \tilde{x}_{t-1}\| + (1 - \mu\eta_{y_t}/2)\|y_{t-1} - \tilde{y}_{t-1}\|.$$

$\square$

**Lemma 11.** *With the update rules defined in Algorithm 1, it holds that*

$$\mathbb{E}\|m_t - \tilde{m}_t\| \le \mathbb{E}\left[(1 - \eta_{m_t})\|m_{t-1} - \tilde{m}_{t-1}\| + \eta_m C_m\left(\|x_{t-1} - \tilde{x}_{t-1}\| + \|y_{t-1} + \tilde{y}_{t-1}\|\right)\right] + \eta_{m_t}\left(\frac{2L_0}{n} + \frac{2L_1 L_0}{n\mu}\right),$$

*where $C_m = \frac{2(n-1)L_1}{n} + 2L_2 D_z + \frac{L_1}{\mu}\left(\frac{(n-1)L_1}{n} + D_z L_2\right)$.*

*Proof.* With the update rules, it holds that

$$\mathbb{E}\|m_t - \tilde{m}_t\|$$
$$= \mathbb{E}\|(1 - \eta_{m_t})m_{t-1} + \eta_{m_t}\left(\nabla_x f\left(x_{t-1}, y_{t-1}, \xi_t^{(1)}\right) - \nabla_{xy}^2 g\left(x_{t-1}, y_{t-1}, \zeta_t^{(K+2)}\right)z_t^K\right)$$
$$\quad - (1 - \eta_{m_t})\tilde{m}_{t-1} - \eta_{m_t}\left(\nabla_x f\left(\tilde{x}_{t-1}, \tilde{y}_{t-1}, \tilde{\xi}_t^{(1)}\right) - \nabla_{xy}^2 g\left(\tilde{x}_{t-1}, y_{t-1}, \zeta_t^{(K+2)}\right)\tilde{z}_t^K\right)\|$$
$$\le \mathbb{E}\left[(1 - \eta_{m_t})\|m_{t-1} - \tilde{m}_{t-1}\| + \eta_{m_t}\|\nabla_x f\left(x_{t-1}, y_{t-1}, \xi_t^{(1)}\right) - \nabla_x f\left(\tilde{x}_{t-1}, \tilde{y}_{t-1}, \tilde{\xi}_t^{(1)}\right)\|\right.$$
$$\left. + \eta_{m_t}\|\nabla_{xy}^2 g\left(x_{t-1}, y_{t-1}, \zeta_t^{(K+2)}\right)z_t^K - \nabla_{xy}^2 g\left(\tilde{x}_{t-1}, \tilde{y}_{t-1}, \zeta_t^{(K+2)}\right)\tilde{z}_t^K\|\right]$$

On the one hand, when $\tilde{\xi}_t^{(1)} \ne \xi_t^{(1)}$, it holds that $\left\|\nabla_x f\left(x_{t-1}, y_{t-1}, \xi_t^{(1)}\right) - \nabla_x f\left(\tilde{x}_{t-1}, \tilde{y}_{t-1}, \tilde{\xi}_t^{(1)}\right)\right\| \le 2L_0$. When $\tilde{\xi}_t^{(1)} = \xi_t^{(1)}$, it holds that $\left\|\nabla_x f\left(x_{t-1}, y_{t-1}, \xi_t^{(1)}\right) - \nabla_x f\left(\tilde{x}_{t-1}, \tilde{y}_{t-1}, \tilde{\xi}_t^{(1)}\right)\right\| \le 2L_1\left(\|x_{t-1} - \tilde{x}_{t-1}\| + \|y_{t-1} - \tilde{y}_{t-1}\|\right)$.

Meanwhile, $\tilde{\xi}_t^{(1)} \ne \xi_t^{(1)}$ with probability $1/n$, while $\tilde{\xi}_t^{(1)} = \xi_t^{(1)}$ with probability $1 - 1/n$.

Thus, $\mathbb{E}\|\nabla_x f\left(x_{t-1}, y_{t-1}, \xi_t^{(1)}\right) - \nabla_x f\left(\tilde{x}_{t-1}, \tilde{y}_{t-1}, \tilde{\xi}_t^{(1)}\right)\| \le \frac{2L_0}{n} + 2\left(1 - \frac{1}{n}\right)L_1\mathbb{E}\left[\|x_{t-1} - \tilde{x}_{t-1}\| + \|y_{t-1} - \tilde{y}_{t-1}\|\right]$.

On the other hand, it holds that

$$\mathbb{E}\|\nabla_{xy}^2 g\left(x_{t-1}, y_{t-1}, \zeta_t^{(K+2)}\right)z_t^K - \nabla_{xy}^2 g\left(\tilde{x}_{t-1}, \tilde{y}_{t-1}, \zeta_t^{(K+2)}\right)\tilde{z}_t^K\|$$
$$\le \mathbb{E}\left[\|\nabla_{xy}^2 g\left(x_{t-1}, y_{t-1}, \zeta_t^{(K+1)}\right) - \nabla_{xy}^2 g\left(\tilde{x}_{t-1}, \tilde{y}_{t-1}, \zeta_t^{(K+1)}\right)\|\|z_t^K\| + \|z_t^K - \tilde{z}_t^K\|\|\nabla_{xy}^2 g\left(\tilde{x}_{t-1}, \tilde{y}_{t-1}, \zeta_t^{(K+2)}\right)\|\right]$$
$$\le \mathbb{E}\left[2L_2 D_z\left(\|x_{t-1} - \tilde{x}_{t-1}\| - \|y_{t-1} - \tilde{y}_{t-1}\|\right) + \frac{L_1}{\mu}\left(\frac{(n-1)L}{n} + D_z L_2\right)\left(\|x_{t-1} - \tilde{x}_{t-1}\| + \|y_{t-1} - \tilde{y}_{t-1}\|\right)\right.$$
$$\left. + \frac{2L_1 L_0}{n\mu}\right]$$
$$\le \mathbb{E}\left[\left(2L_2 D_z + \frac{L_1}{\mu}\left(\frac{(n-1)L_1}{n} + D_z L_2\right)\right)\left(\|x_{t-1} - \tilde{x}_{t-1}\| + \|y_{t-1} - \tilde{y}_{t-1}\|\right) + \frac{2L_1 L_0}{n\mu}\right],$$

where the second inequality is based on Lemma 9.

Therefore, by combining the above inequalities, it holds that

$$\mathbb{E}\|m_t - \tilde{m}_t\| \le \mathbb{E}\left[(1-\eta_{m_t})\|m_{t-1} - \tilde{m}_{t-1}\| + 2\eta_{m_t}\left(1 - \frac{1}{n}\right)L_1\left(\|x_{t-1} - \tilde{x}_{t-1}\| + \|y_{t-1} - \tilde{y}_{t-1}\|\right) + \frac{2\eta_{m_t}L_0}{n}\right.$$

$$\left. + \eta_{m_t}\left(2L_2 D_z + \frac{L_1}{\mu}\left(\frac{(n-1)L_1}{n} + D_z L_2\right)\right)\left(\|x_{t-1} - \tilde{x}_{t-1}\| + \|y_{t-1} - \tilde{y}_{t-1}\|\right) + \frac{2L_1 L_0 \eta_{m_t}}{n\mu}\right]$$

By defining $C_m = \frac{2(n-1)L_1}{n} + 2L_2 D_z + \frac{L_1}{\mu}\left(\frac{(n-1)L_1}{n} + D_z L_2\right)$, we get the desired result. $\qquad\square$

**Lemma 12.** *With the update rules defined in Algorithm 1, it holds that*
$$\mathbb{E}\|x_t - \tilde{x}_t\|$$
$$\le \mathbb{E}\left[(1 + \eta_{x_t}\eta_{m_t}C_m)\|x_{t-1} - \tilde{x}_{t-1}\| + \eta_{x_t}\eta_{m_t}C_m\|y_{t-1} - \tilde{y}_{t-1}\| + \eta_{x_t}(1 - \eta_{m_t})\|m_{t-1} - \tilde{m}_{t-1}\|\right]$$
$$+ \eta_{x_t}\eta_{m_t}\left(\frac{2L_0}{n} + \frac{2L_1 L_0}{n\mu}\right),$$
*where $C_m = \frac{2(n-1)L_1}{n} + 2L_2 D_z + \frac{L_1}{\mu}\left(\frac{(n-1)L_1}{n} + D_z L_2\right)$*

*Proof.* With the update rules defined in Algorithm 1, it holds that
$$\mathbb{E}\|x_t - \tilde{x}_t\| = \mathbb{E}\|x_{t-1} - \eta_{x_t}m_t - \tilde{x}_{t-1} + \eta_{x_t}\tilde{m}_t\|$$
$$\le \mathbb{E}\left[\|x_{t-1} - \tilde{x}_{t-1}\|\right] + \eta_{x_t}\|m_t - \tilde{m}_t\|$$
$$\le \mathbb{E}\left[(1 + \eta_{x_t}\eta_{m_t}C_m)\|x_{t-1} - \tilde{x}_{t-1}\| + \eta_{x_t}\eta_{m_t}C_m\|y_{t-1} - \tilde{y}_{t-1}\| + \eta_{x_t}(1 - \eta_{m_t})\|m_{t-1} - \tilde{m}_{t-1}\|\right]$$
$$+ \eta_{x_t}\eta_{m_t}\left(\frac{2L_0}{n} + \frac{2L_1 L_0}{\mu n}\right).$$
where the last inequality is based on Lemma 11. $\qquad\square$

*Proof of Theorem 1.* Based on Lemma 10,11 and 12, it holds that
$$\mathbb{E}\left[\|x_t - \tilde{x}_t\| + \|y_t - \tilde{y}_t\| + \|m_t - \tilde{m}_t\|\right]$$
$$\le \mathbb{E}\left[(1 + \eta_{x_t}\eta_{m_t}C_m + \eta_{m_t}C_m + \eta_{y_t}L_1)\|x_{t-1} - \tilde{x}_{t-1}\|\right.$$
$$+ (1 - \mu\eta_y/2 + \eta_{m_t}C_m + \eta_{x_t}\eta_{m_t}C_m)\|y_{t-1} - \tilde{y}_{t-1}\|$$
$$+ (1 + \eta_{x_t})(1 - \eta_{m_t})\|m_{t-1} - \tilde{m}_{t-1}\|\right] + (1 + \eta_{x_t})\eta_{m_t}\left(\frac{2L_0}{n} + \frac{2L_1 L_0}{n\mu}\right)$$
$$\le \mathbb{E}\left[(1 + \eta_{x_t}\eta_{m_t}C_m + \eta_{m_t}C_m + \eta_{y_t}L_1)\left(\|x_{t-1} - \tilde{x}_{t-1}\| + \|y_{t-1} - \tilde{y}_{t-1}\| + \|m_{t-1} + \tilde{m}_{t-1}\|\right)\right]$$
$$+ (1 + \eta_{x_t})\eta_{m_t}\left(\frac{2L_0}{n} + \frac{2L_1 L_0}{n\mu}\right)$$

Thus, by induction, we get the desired result. $\qquad\square$

*Proof of Corollary 1.* According to Theorem 1, when $\eta_{x_t} = \eta_{m_t} = \alpha/t$ and $\eta_{y_t} = \beta/t$, it holds that
$$\mathbb{E}\left[\|x_t - \tilde{x}_t\| + \|y_t - \tilde{y}_t\| + \|m_t - \tilde{m}_t\|\right]$$
$$\le (1 + (2C_m\alpha + L_1\beta)/t)\mathbb{E}\left[\|x_{t-1} - \tilde{x}_{t-1}\| + \|y_{t-1} - \tilde{y}_{t-1}\| + \|m_{t-1} - \tilde{m}_{t-1}\|\right] + (1 + \eta_{x_t})\eta_{m_t}\left(\frac{2L_0}{n} + \frac{2L_1 L_0}{n\mu}\right)$$
$$\le exp\left((2C_m\alpha + L_1\beta)/t\right)\mathbb{E}\left[\|x_{t-1} - \tilde{x}_{t-1}\| + \|y_{t-1} - \tilde{y}_{t-1}\| + \|m_{t-1} - \tilde{m}_{t-1}\|\right] + \frac{4L_0}{nt} + \frac{4L_1 L_0}{nt\mu}$$

Thus, it holds that
$$\mathbb{E}\left[\|x_T - \tilde{x}_T\| + \|y_T - \tilde{y}_T\| + \|m_T - \tilde{m}_T\|\right.$$
$$\left.\mid \|x_{t_0} - \tilde{x}_{t_0}\| + \|y_{t_0} - \tilde{y}_{t_0}\| + \|m_{t_0} - \tilde{m}_{t_0}\| = 0\right]$$
$$\le \sum_{t=t_0+1}^{T} exp\left((2C_m\alpha + L_1\beta)\log\left(\frac{T}{t}\right)\right)\left(\frac{4L_0}{nt} + \frac{4L_1 L_0}{nt\mu}\right)$$
$$\le \left(\frac{4L_0}{n(2C_m\alpha + L_1\beta)} + \frac{4L_1 L_0}{n\mu(2C_m\alpha + L_1\beta)}\right)\left(\frac{T}{t_0}\right)^{2C_m\alpha + L_1\beta}$$

When $f(x, y, \xi) \in [0, 1]$, it holds that

$$|f(x_T, y_T; \xi) - f(\tilde{x}_T, \tilde{y}_T, \xi)| \leq \frac{t_0}{n} + L_0 \mathbb{E}\left[\|x_T - \tilde{x}_T\| + \|y_T - \tilde{y}_T\| \mid \|x_{t_0} - \tilde{x}_{t_0}\| + \|y_{t_0} - \tilde{y}_{t_0}\| = 0\right]$$

Combining the above inequalities, it holds that

$$|f(x_T, y_T; \xi) - f(\tilde{x}_T, \tilde{y}_T, \xi)| \leq \frac{t_0}{n} + \frac{4L_0}{n(2C_m\alpha + L_1\beta)} + \frac{4L_1L_0}{n\mu(2C_m\alpha + L_1\beta)}\left(\frac{T}{t_0}\right)^{2C_m\alpha + L_1\beta}$$

Thus, by choosing $t_0 = \left(\frac{4L_0}{2C_m\alpha + L_1\beta} + \frac{4L_1L_0}{2\mu C_m\alpha + \mu L_1\beta}\right)^{\frac{1}{2C_m\alpha + L_1\beta + 1}} T^{\frac{2C_m\alpha + L_1\beta}{2C_m\alpha + L_1\beta + 1}}$, it holds that

$$|f(x_T, y_T; \xi) - f(\tilde{x}_T, \tilde{y}_T, \xi)| \leq \frac{1}{n}\left(1 + \left(\frac{4L_0}{2C_m\alpha + L_1\beta} + \frac{4L_1L_0}{2C_m\mu\alpha + L_1\mu\beta}\right)^{\frac{1}{2C_m\alpha + L_1\beta + 1}}\right) T^{\frac{2C_m\alpha + L_1\beta}{2C_m\alpha + L_1\beta + 1}}$$

$$= \mathcal{O}\left(\frac{T^q}{n}\right)$$

where $q = \frac{2C_m\alpha + L_1\beta}{2C_m\alpha + L_1\beta + 1} < 1$

$\square$

## B  PROOF OF THEOREM 2

Define $\Phi(x) = \frac{1}{n}\sum_{i=1}^{n} f(x, y^*(x), \xi_i)$

**Lemma 13.** *Based on definition of $\Phi$, it holds that*

$$\nabla\Phi(x) = \frac{1}{n}\sum_{i=1}^{n}\nabla_x f(x, y^*(x), \xi_i)$$

$$- \frac{1}{nq}\left(\sum_{j=1}^{q}\nabla_{xy}^2 g(x, y^*(x), \zeta_j)\right)\left(\frac{1}{q}\sum_{j=1}^{q}\nabla_{yy}^2 g(x, y^*(x), \zeta_j)\right)^{-1}\left(\sum_{i=1}^{n}\nabla_y f(x, y^*(x), \xi_i)\right)$$

*Proof.* By the chain rule, it holds that

$$\Phi(x) = \frac{1}{n}\sum_{i=1}^{n}\nabla_x f(x, y^*(x), \xi_i) - \frac{\partial y^*(x)}{\partial x}\frac{1}{n}\sum_{i=1}^{n}\nabla_y f(x, y^*(x), \xi_i).$$

With the optimality condition of $y^*(x)$, it holds that

$$\frac{1}{m}\sum_{j=1}^{q}\nabla_y g(x, y^*(x), \zeta_j) = 0.$$

Taking the gradient of $x$ on both sides of the equation, it holds that

$$\frac{1}{q}\sum_{j=1}^{q}\nabla_{xy}^2 g(x, y^*(x), \zeta_j) + \frac{\partial y^*(x)}{\partial x}\frac{1}{q}\sum_{j=1}^{q}\nabla_{yy}^2 g(x, y^*(x), \zeta_j) = 0.$$

Thus, it holds that

$$\frac{\partial y^*(x)}{\partial x} = -\left(\frac{1}{q}\sum_{j=1}^{q}\nabla_{xy}^2 g(x, y^*(x), \zeta_j)\right)\left(\frac{1}{q}\sum_{j=1}^{q}\nabla_{yy}^2 g(x, y^*(x), \zeta_j)\right)^{-1}$$

Thus, we give the desired result. $\square$

**Lemma 14.** *By the definition of $y^*(x)$, it holds that $\|y^*(x_1) - y^*(x_2)\| \leq L_y\|x_1 - x_2\|$, where* $L_y = \frac{L_1}{\mu}$

*Proof.* It holds that

$$\left\| \frac{\partial y^* (x)}{\partial x} \right\|_2$$

$$= \left\| - \left( \frac{1}{q} \sum_{j=1}^q \nabla_{xy}^2 g (x, y^* (x), \zeta_j) \right) \left( \frac{1}{q} \sum_{j=1}^q \nabla_{yy}^2 g (x, y^* (x), \zeta_j) \right)^{-1} \right\|$$

$$\leq \left\| \frac{1}{q} \sum_{j=1}^q \nabla_{xy}^2 g (x, y^* (x), \zeta_j) \right\| \left\| \left( \frac{1}{q} \sum_{j=1}^q \nabla_{yy}^2 g (x, y^* (x), \zeta_j) \right)^{-1} \right\|$$

$$\leq \frac{L_1}{\mu}$$

Thus, by the fundamental theorem of calculus, we can obtain that

$$\| y^* (x_1) - y^* (x_2) \| \leq \| \frac{\partial y^* (z)}{\partial z} \|_2 \| x_1 - x_2 \| \leq \frac{L_1}{\mu} \| x_1 - x_2 \|$$

where $z$ lies in the segment $[x_1, x_2]$. □

**Lemma 15.** *The gradients of $\Phi$ are Lipschitz with Lipschitz constant $L_\Phi$ = $\frac{(1 + L_y) (L_1 \mu^2 + L_0 L_2 \mu + L_1^2 \mu + L_2 L_0)}{\mu^2}$*

*Proof.* According to lemma 13, it holds that
$$\| \nabla \Phi (x_1) - \nabla \Phi (x_2) \|$$

$$= \| \nabla_x \frac{1}{n} \sum_{i=1}^n f (x_1, y^* (x_1), \xi_i)$$

$$- \frac{1}{nq} \left( \sum_{j=1}^q \nabla_{xy}^2 g (x_1, y^* (x_1), \zeta_j) \right) \left( \frac{1}{q} \sum_{j=1}^q \nabla_{yy}^2 g (x_1, y^* (x_1), \zeta_j) \right)^{-1} \left( \nabla_y \sum_{i=1}^n f (x_1, y^* (x_1), \xi_i) \right)$$

$$- \nabla_x \frac{1}{n} \sum_{i=1}^n f (x_2, y^* (x_2), \xi_i)$$

$$+ \frac{1}{nq} \left( \sum_{j=1}^q \nabla_{xy}^2 g (x_2, y^* (x_2), \zeta_j) \right) \left( \frac{1}{q} \sum_{j=1}^q \nabla_{yy}^2 g (x_2, y^* (x_2), \zeta_j) \right)^{-1} \left( \sum_{i=1}^n \nabla_y f (x_2, y^* (x_2), \xi_i) \right) \|$$

$$\leq \frac{1}{n} \sum_{i=1}^n \| \nabla_x f (x_1, y^* (x_1), \xi_i) - \nabla_x f (x_2, y^* (x_2), \xi_i) \|$$

$$+ \frac{1}{q} \sum_{j=1}^q \| \nabla_{xy}^2 g (x_1, y^* (x_1), \zeta_j) - \nabla_{xy}^2 g (x_2, y^* (x_2), \zeta_j) \| \left\| \left( \frac{1}{q} \sum_{j=1}^q \nabla_{yy}^2 g (x_1, y^* (x_1), \zeta_j) \right)^{-1} \right\| \| \frac{1}{n} \nabla_y \sum_{i=1}^n \nabla_y f (x_1, y^* (x_1), \xi_i) \|$$

$$+ \| \frac{1}{q} \sum_{j=1}^q \nabla_{xy}^2 g (x_2, y^* (x_2), \zeta_j) \| \left\| \left( \frac{1}{q} \sum_{j=1}^q \nabla_{yy}^2 g (x_1, y^* (x_1), \zeta_j) \right)^{-1} - \left( \frac{1}{q} \sum_{j=1}^q \nabla_{yy}^2 g (x_2, y^* (x_2), \zeta_j) \right)^{-1} \right\|$$

$$\| \frac{1}{n} \sum_{i=1}^n \nabla_y f (x_1, y^* (x_1), \xi_i) \|$$

$$+ \frac{1}{n} \sum_{i=1}^n \| \frac{1}{q} \sum_{j=1}^q \nabla_{xy}^2 g (x_2, y^* (x_2), \zeta_j) \| \left\| \left( \frac{1}{q} \sum_{j=1}^q \nabla_{yy}^2 g (x_2, y^* (x_2), \zeta_j) \right)^{-1} \right\| \| \nabla_y f (x_1, y^* (x_1), \xi_i) - \nabla_y f (x_2, y^* (x_2), \xi_i) \|$$

$$\leq L_1 (\| x_1 - x_2 \| + \| y^* (x_1) - y^* (x_2) \|) + \frac{L_2 L_0}{\mu} (\| x_1 - x_2 \| + \| y^* (x_1) - y^* (x_2) \|)$$

$$+ \frac{L_2 L_0}{\mu^2} (\| x_1 - x_2 \| + \| y^* (x_1) - y^* (x_2) \|) + \frac{L_1^2}{\mu} (\| x_1 - x_2 \| + \| y^* (x_1) - y^* (x_2) \|)$$

$$\leq \frac{(1 + L_y) (L_1 \mu^2 + L_0 L_2 \mu + L_1^2 \mu + L_2 L_0)}{\mu^2} \| x_1 - x_2 \|$$

where the third inequality is because $\|A^{-1} - B^{-1}\| \leq \|A^{-1}\|\|A - B\|\|B^{-1}\|$.

Hence, we obtain the desired result. $\qquad\square$

**Lemma 16.** *Denote* $\Delta_t = \nabla_x f\left(x_{t-1}, y_{t-1}, \xi_t^{(1)}\right) - \nabla_{xy}^2 g\left(x_{t-1}, y_{t-1}, \zeta_t^{(K+2)}\right) z_t^K$, *then it holds that*

$$\mathbb{E}\|\mathbb{E}\Delta_t - \nabla\Phi\left(x_{t-1}\right)\|^2 \leq 2\left(L_1 + D_z L_2\right)^2 \|y_{t-1} - y^*\left(x_{t-1}\right)\|^2 + 2L_1^2\left(1 - \eta_z\mu\right)^{2K}\left(D_z + \frac{L_0}{\mu}\right)^2$$

*Proof.* By the update rules, it holds that

$$z_t^K = \left(I - \eta_z\nabla_{yy}^2 g\left(x_{t-1}, y_{t-1}, \zeta_t^{(1)}\right)\right) z_t^{K-1} + \eta_y\nabla_y f\left(x_{t-1}, y_{t-1}, \xi_t^{(1)}\right)$$
$$= \cdots$$
$$= \sum_{k=1}^{K}\eta_z\Pi_{t=k+1}^{K}\left(I - \eta_z\nabla_{yy}^2 g\left(x_{t-1}, y_{t-1}, \zeta_t^{(t)}\right)\right)\nabla_y f\left(x_{t-1}, y_{t-1}, \zeta_t^{(1)}\right) + \Pi_{k=1}^{K}\left(I - \eta_z\nabla_{yy}g\left(x_{t-1}, y_{t-1}, \zeta_t^{(k)}\right)\right) z_0$$

Thus, it holds that

$$\mathbb{E}z_t^K$$
$$= \mathbb{E}\left[\eta_z\sum_{k=0}^{K-1}\left(I - \eta_z\left(\frac{1}{q}\sum_{j=1}^{q}\nabla_{yy}^2 g\left(x_{t-1}, y_{t-1}, \zeta_j\right)\right)\right)^k\left(\frac{1}{n}\sum_{i=1}^{n}\nabla_y f\left(x_{t-1}, y_{t-1}, \xi_i\right)\right)\right.$$
$$\left. + \left(I - \eta_z\left(\frac{1}{q}\sum_{j=1}^{q}\nabla_{yy}^2 g\left(x_{t-1}, y_{t-1}, \zeta_j\right)\right)\right)^K z_0\right]$$

Hence, we can obtain that

$$\|\mathbb{E}\Delta_t - \nabla\Phi\left(x_{t-1}\right)\|$$
$$\leq \frac{1}{n}\sum_{i=1}^{n}\|\nabla_x f\left(x_{t-1}, y_{t-1}, \xi_i\right) - \nabla_x f\left(x_{t-1}, y^*\left(x_{t-1}\right), \xi_i\right)\|$$
$$\quad + \frac{1}{q}\|\sum_{j=1}^{q}\nabla_{xy}^2 g\left(x_{t-1}, y_{t-1}, \zeta_j\right) - \nabla_{xy}^2 g\left(x_{t-1}, y^*\left(x_{t-1}\right), \zeta_j\right)\|\|\mathbb{E}z_t^K\|$$
$$\quad + \|\frac{1}{q}\sum_{j=1}^{q}\nabla_{xy}^2 g\left(x_{t-1}, y^*\left(x_{t-1}\right), \zeta_j\right)\|$$
$$\quad \left\|\mathbb{E}z_t^K - \left(\frac{1}{q}\sum_{j=1}^{q}\nabla_{yy}^2 g\left(x_{t-1}, y^*\left(x_{t-1}\right), \zeta_j\right)\right)^{-1}\left(\frac{1}{n}\sum_{i=1}^{n}f\left(x_{t-1}, y^*\left(x_{t-1}\right), \xi_i\right)\right)\right\|$$
$$\leq L_1\|y_{t-1} - y^*\left(x_{t-1}\right)\| + D_z L_2\|y_{t-1} - y^*\left(x_{t-1}\right)\|$$
$$\quad + L_1\left\|\mathbb{E}z_t^K - \left(\frac{1}{q}\sum_{j=1}^{q}\nabla_{yy}^2 g\left(x_{t-1}, y^*\left(x_{t-1}\right), \zeta_j\right)\right)^{-1}\left(\frac{1}{n}\sum_{i=1}^{n}\nabla_y f\left(x_{t-1}, y^*\left(x_{t-1}\right), \xi_i\right)\right)\right\|$$

Meanwhile, it holds that

$$
\left\| \mathbb{E} z_t^K - \left( \frac{1}{q} \sum_{j=1}^q \nabla_{yy}^2 g\left(x_{t-1}, y^*\left(x_{t-1}\right), \zeta_j\right) \right)^{-1} \left( \frac{1}{n} \sum_{i=1}^n f\left(x_{t-1}, y^*\left(x_{t-1}\right), \xi_i\right) \right) \right\|
$$

$$
\leq \left\| \mathbb{E} \left[ \eta_z \sum_{k=0}^K \left( I - \eta_z \left( \frac{1}{q} \sum_{j=1}^q \nabla_{yy}^2 g\left(x_{t-1}, y_{t-1}, \zeta_j\right) \right) \right)^k \left( \frac{1}{n} \sum_{i=1}^n \nabla_y f\left(x_{t-1}, y_{t-1}, \xi_i\right) \right) \right] \right.
$$

$$
\left. - \left( \frac{1}{q} \sum_{j=1}^q \nabla_{yy}^2 g\left(x_{t-1}, y^*\left(x_{t-1}\right), \zeta_j\right) \right)^{-1} \left( \frac{1}{n} \sum_{i=1}^n \nabla_y f\left(x_{t-1}, y^*\left(x_{t-1}\right), \xi_i\right) \right) \right\|
$$

$$
+ \left\| \left( I - \eta_z \left( \frac{1}{q} \sum_{j=1}^q \nabla_{yy}^2 g\left(x_{t-1}, y_{t-1}, \zeta_j\right) \right) \right)^K z_0 \right\|
$$

$$
\leq \left(1 - \eta_z \mu\right)^K D_z + \frac{L_0 \left(1 - \eta_z \mu\right)^K}{\mu}
$$

Thus, it holds that

$$
\left\| \mathbb{E} \Delta_t - \nabla \Phi\left(x_{t-1}\right) \right\|^2 \leq 2 \left(L_1 + D_z L_2\right)^2 \left\| y_{t-1} - y^*\left(x_{t-1}\right) \right\|^2 + 2 L_1^2 \left(1 - \eta_z \mu\right)^{2K} \left( D_z + \frac{L_0}{\mu} \right)^2
$$

$\square$

**Lemma 17.** *Denote* $\Delta_t = \nabla_x f\left(x_{t-1}, y_{t-1}, \xi_t^{(1)}\right) - \nabla_{xy}^2 g\left(x_{t-1}, y_{t-1}, \zeta_t^{(K+2)}\right) z_t^K$, *then it holds that*

$$
\mathbb{E}\|\Delta_t - \mathbb{E}\Delta_t\|^2 \leq L_0^2 + 2 L_1^2 \left( K L_1^2 D_z^2 + 2K \eta_z^2 L_1^2 L_0^2 + \frac{2K L_0^2}{\mu^2} \right) + 2 D_z^2 L_1^2
$$

*Proof.* With the definition of $\Delta_t$, and the calculation of $\mathbb{E}\Delta_t$, it holds that

$$
\mathbb{E}\|\Delta_t - \mathbb{E}\Delta_t\|^2
$$

$$
= \mathbb{E} \left\| \nabla_x f\left(x_{t-1}, y_{t-1}, \xi_t^{(1)}\right) - \nabla_{xy}^2 g\left(x_{t-1}, y_{t-1}, \zeta_t^{(K+2)}\right) z_t^K \right.
$$

$$
- \left[ \frac{1}{n} \sum_{i=1}^n \nabla_x f\left(x_{t-1}, y_{t-1}, \xi_i\right) - \left( I - \eta_z \left( \frac{1}{q} \sum_{j=1}^q \nabla_{yy}^2 g\left(x_{t-1}, y_{t-1}, \zeta_j\right) \right) \right)^K z_0 \right.
$$

$$
\left. \left. - \eta_z \sum_{k=0}^{K-1} \left( I - \eta_z \left( \frac{1}{q} \sum_{j=1}^q \nabla_{yy}^2 g\left(x_{t-1}, y_{t-1}, \zeta_j\right) \right) \right)^k \left( \frac{1}{n} \sum_{i=1}^n \nabla_y f\left(x_{t-1}, y_{t-1}, \xi_i\right) \right) \right] \right\|^2
$$

$$
= \mathbb{E} \left\| \nabla_x f\left(x_{t-1}, y_{t-1}, \xi_t^{(1)}\right) - \frac{1}{n} \sum_{i=1}^n \nabla_x f\left(x_{t-1}, y_{t-1}, \xi_i\right) \right\|^2
$$

$$
+ \mathbb{E} \left\| \nabla_{xy}^2 g\left(x_{t-1}, y_{t-1}, \zeta_t^{(K+2)}\right) z_t^K - \left( I - \eta_z \left( \frac{1}{q} \sum_{j=1}^q \nabla_{yy}^2 g\left(x_{t-1}, y_{t-1}, \zeta_j\right) \right) \right)^K z_0 \right.
$$

$$
\left. - \eta_z \sum_{k=0}^{K-1} \left( I - \eta_z \left( \frac{1}{q} \sum_{j=1}^q \nabla_{yy}^2 g\left(x_{t-1}, y_{t-1}, \zeta_j\right) \right) \right)^k \left( \frac{1}{n} \sum_{i=1}^n \nabla_y f\left(x_{t-1}, y_{t-1}, \xi_i\right) \right) \right\|^2
$$

$$
\leq L_0^2 + 2 L_1^2 \mathbb{E}\|z_t^K - \mathbb{E} z_t^K\|^2 + 2 D_z^2 L_1^2
$$

Meanwhile, it holds that

$$\mathbb{E}\|z_t^K - \mathbb{E}z_t^K\|^2$$

$$= \mathbb{E}\|\sum_{k=1}^{K}\eta_z\Pi_{t=k+1}^{K}\left(I - \eta_z\nabla_{yy}^2 g\left(x_{t-1}, y_{t-1}, \zeta_t^{(t)}\right)\right)\nabla_y f\left(x_{t-1}, y_{t-1}, \xi_t^{(1)}\right) + \Pi_{k=1}^{K}\left(I - \eta_z\nabla g\left(x_{t-1}, y_{t-1}, \zeta_t^{(k)}\right)\right)z_0$$

$$- \left[\eta_z\sum_{k=0}^{K-1}\left(I - \eta_z\left(\frac{1}{q}\sum_{j=1}^{q}\nabla_{yy}^2 g\left(x_{t-1}, y_{t-1}, \zeta_j\right)\right)\right)^k\left(\frac{1}{n}\sum_{i=1}^{n}\nabla_y f\left(x_{t-1}, y_{t-1}, \xi_i\right)\right)\right.$$

$$\left. + \left(I - \eta_z\left(\frac{1}{q}\sum_{j=1}^{q}\nabla_{yy}^2 g\left(x_{t-1}, y_{t-1}, \zeta_j\right)\right)\right)^K z_0\right]\|^2$$

$$\leq K\sum_{k=0}^{K-1}\left\|\eta_z\left[\left(I - \eta_z\left(\frac{1}{q}\sum_{j=1}^{q}\nabla_{yy}^2 g\left(x_{t-1}, y_{t-1}, \zeta_j\right)\right)\right)^k - \Pi_{t=K-k+1}^{K}\left(I - \eta_z\nabla_{yy}^2 g\left(x_{t-1}, y_{t-1}, \zeta_t^{(t)}\right)\right)\right]\right.$$

$$\left.\left(\frac{1}{n}\sum_{i=1}^{n}\nabla_y f\left(x_{t-1}, y_{t-1}, \xi_i\right)\right)\right\|^2$$

$$+ K\left\|\left[\Pi_{k=1}^{K}\left(I - \eta_z\nabla g\left(x_{t-1}, y_{t-1}, \zeta_t^{(k)}\right)\right) - \left(I - \eta_z\left(\frac{1}{q}\sum_{j=1}^{q}\nabla_{yy}^2 g\left(x_{t-1}, y_{t-1}, \zeta_j\right)\right)\right)^K\right]z_0\right\|^2$$

$$\leq KL_1^2 D_z^2 + 2K\eta_z^2 L_1^2 L_0^2 + \frac{2KL_0^2}{\mu^2}$$

Thus, plugging the bounded of $\mathbb{E}\|z_t^K - \mathbb{E}z_t^K\|^2$ into the above inequality, we obtain the desired result $\qquad\square$

**Lemma 18.** *According to the update rules, it holds that*

$$\|m_t\|^2 \leq 2L_0^2 + 2L_1^2 D_z^2$$
$$\|\Delta_t\|^2 \leq 2L_0^2 + 2L_1^2 D_z^2$$

*Proof.* By the definition of $\Delta_t$, it holds that

$$\|\Delta_t\|^2 = \|\nabla_x f\left(x_{t-1}, y_{t-1}, \xi_t^{(1)}\right) - \nabla_{xy}^2 g\left(x_{t-1}, y_{t-1}, \zeta^{(K+2)}\right)z_t^K\|^2 \leq 2L_0^2 + 2L_1^2 D_z^2$$

By the definition of $m_t$, it holds that

$$\|m_t\| \leq (1 - \eta_{m_t})\|m_{t-1}\| + \eta_{m_t}\|\Delta_t\| \leq L_0 + L_1 D_z$$

Thus, it holds that

$$\|m_t\|^2 \leq 2L_0^2 + 2L_1^2 D_z^2.$$

$\qquad\square$

**Lemma 19.** *With the update rules of $m_t$, it holds that*

$$\sum_{t=1}^{T}\eta_{m_{t+1}}\mathbb{E}\|m_t - \nabla\Phi\left(x_t\right)\|^2$$

$$\leq \mathbb{E}\|\nabla\Phi\left(x_0\right)\|^2 + \sum_{t=1}^{T}2\eta_{m_t}\mathbb{E}\|\mathbb{E}\Delta_t - \nabla\Phi\left(x_{t-1}\right)\|^2 + 2\eta_{x_t}^2/\eta_{m_t}L_1^2\|m_t\|^2 + \eta_{m_t}^2\mathbb{E}\|\Delta_t - \mathbb{E}\Delta_t\|^2.$$

*Proof.* With the update rules of $m_t$, it holds that

$$\mathbb{E}\|m_t - \nabla\Phi(x_t)\|^2$$

$$= \mathbb{E}\|(1-\eta_{m_t})m_{t-1} + \eta_{m_t}\Delta_t - \nabla\Phi(x_{t-1}) + \nabla\Phi(x_{t-1}) - \nabla\Phi(x_t)\|^2$$

$$= \mathbb{E}\|(1-\eta_{m_t})m_{t-1} + \eta_{m_t}\mathbb{E}\Delta_t - \nabla\Phi(x_{t-1}) + \nabla\Phi(x_{t-1}) - \nabla\Phi(x_t)\|^2 + \eta_{m_t}^2\mathbb{E}\|\Delta_t - \mathbb{E}\Delta_t\|^2$$

$$= (1-\eta_{m_t})\mathbb{E}\|m_{t-1} - \nabla\Phi(x_{t-1})\|^2 + \eta_{m_t}\|\mathbb{E}\Delta_t - \nabla\Phi(x_{t-1}) + 1/\eta_{m_t}(\nabla\Phi(x_{t-1}) - \nabla\Phi(x_t))\|^2$$
$$\qquad + \eta_{m_t}^2\mathbb{E}\|\Delta_t - \mathbb{E}\Delta_t\|^2$$

$$\leq (1-\eta_{m_t})\mathbb{E}\|m_{t-1} - \nabla\Phi(x_{t-1})\|^2 + 2\eta_{m_t}\|\mathbb{E}\Delta_t - \nabla\Phi(x_{t-1})\|^2 + 2\eta_{x_t}^2/\eta_{m_t}L_1^2\|m_t\|^2$$
$$\qquad + \eta_{m_t}^2\mathbb{E}\|\Delta_t - \mathbb{E}\Delta_t\|^2$$

By summing the above inequality up, it holds that

$$\sum_{t=1}^T \eta_{m_{t+1}}\mathbb{E}\|m_t - \nabla\Phi(x_t)\|^2$$

$$\leq \mathbb{E}\|\nabla\Phi(x_0)\|^2 + \sum_{t=1}^T 2\eta_{m_t}\mathbb{E}\|\mathbb{E}\Delta_t - \nabla\Phi(x_{t-1})\|^2 + 2\eta_{x_t}^2/\eta_{m_t}L_1^2\|m_t\|^2 + \eta_{m_t}^2\mathbb{E}\|\Delta_t - \mathbb{E}\Delta_t\|^2.$$

$$\square$$

**Lemma 20.** *With the update rules of $y_t$ it holds that*

$$\mathbb{E}\|y_t - y^*(x_t)\|^2 \leq (1 - \mu\eta_{y_t}/2)\mathbb{E}\|y_{t-1} - y^*(x_{t-1})\|^2 + \frac{(2+\mu\eta_{y_t})L_1^2\eta_{x_t}^2}{\mu\eta_{y_t}}\mathbb{E}\|m_t\|^2 + 2\eta_{y_t}^2 D_0$$

*Proof.* With the update rules of $y_t$, it holds that

$$\mathbb{E}\|y_t - y^*(x_t)\|^2$$

$$\leq (1+\beta)\mathbb{E}\|y_t - y^*(x_{t-1})\|^2 + (1+1/\beta)\mathbb{E}\|y^*(x_{t-1}) - y^*(x_t)\|^2$$

$$= (1+\beta)\mathbb{E}\|y_{t-1} - \eta_{y_t}\nabla_y g\left(x_{t-1}, y_{t-1}, \zeta^{(K+1)}\right) - y^*(x_{t-1})\|^2 + (1+1/\beta)\mathbb{E}\|y^*(x_{t-1}) - y^*(x_t)\|^2$$

$$\leq (1+\beta)\mathbb{E}\left[ \|y_{t-1} - y^*(x_{t-1})\|^2 - 2\eta_{y_t}\langle y_{t-1} - y^*(x_{t-1}), \frac{1}{q}\sum_{j=1}^q \nabla_y g(x_{t-1}, y_{t-1}, \zeta_j)\rangle \right.$$

$$\left. +\eta_{y_t}^2\|\frac{1}{q}\sum_{j=1}^q \nabla_y g(x_{t-1}, y_{t-1}, \zeta_j)\|^2 + \eta_{y_t}^2\|\frac{1}{q}\sum_{j=1}^q \nabla_y g(x_{t-1}, y_{t-1}, \zeta_j) - \nabla_y g\left(x_{t-1}, y_{t-1}, \zeta_t^{(K+1)}\right)\|^2 \right]$$

$$\quad + (1+1/\beta)L_y^2\eta_{x_t}^2\|m_t\|^2$$

$$\leq (1+\beta)\mathbb{E}\left[ \|y_{t-1} - y^*(x_{t-1})\|^2 - 2\eta_{y_t}\langle y_{t-1} - y^*(x_{t-1}), \frac{1}{q}\sum_{j=1}^q \nabla_y g(x_{t-1}, y_{t-1}, \zeta_j)\rangle \right.$$

$$\left. +\eta_{y_t}^2(1+D_1)\|\frac{1}{q}\sum_{j=1}^q \nabla_y g(x_{t-1}, y_{t-1}, \zeta_j)\|^2 + \eta_{y_t}^2 D_0 \right]$$

$$\quad + (1+1/\beta)L_y^2\eta_{x_t}^2\|m_t\|^2$$

$$\leq (1+\beta)(1-\mu\eta_{y_t})\|y_{t-1} - y^*(x_{t-1})\|^2 + (1+1/\beta)L_y^2\eta_{x_t}^2\|m_{t-1}\|^2 + (1+\beta)\eta_{y_t}^2 D_0.$$

By selecting $\beta = \frac{\mu\eta_{y_t}}{2}$, it holds that

$$\mathbb{E}\|y_t - y^*(x_t)\|^2 \leq \mathbb{E}\left[ (1-\mu\eta_{y_t}/2)\|y_{t-1} - y^*(x_{t-1})\|^2 + \frac{(2+\mu\eta_{y_t})L_1^2\eta_{x_t}^2}{\mu\eta_{y_t}}\|m_t\|^2 \right] + 2\eta_{y_t}^2 D_0.$$

$$\square$$

**Lemma 21.** *With the update rules of $x_t$ and $m_t$, it holds that*

$$\mathbb{E}\left[\frac{\eta_{m_t}}{\eta_{x_t}}\Phi\left(x_t\right) + \frac{1-\eta_{m_t}}{2}\|m_t\|^2 - \frac{\eta_{m_t}}{\eta_{x_t}}\Phi\left(x_{t-1}\right) - \frac{1-\eta_{m_t}}{2}\|m_{t-1}\|^2\right]$$

$$\leq \eta_{m_t}\left(L_1 + D_z L_2\right)^2 \mathbb{E}\|y_{t-1} - y^*\left(x_{t-1}\right)\|^2 + \eta_{m_t} L_1^2 \left(1-\eta_z\mu\right)^{2K}\left(D_z + \frac{L_0}{\mu}\right)^2 - \frac{\eta_{m_t}}{4}\mathbb{E}\|m_t\|^2$$

*Proof.* With the gradient Lipschitz of $\Phi\left(x\right)$, it holds that

$$\mathbb{E}\Phi\left(x_t\right) - \Phi\left(x_{t-1}\right) \leq \mathbb{E}\left[-\eta_{x_t}\langle\nabla\Phi\left(x_{t-1}\right), m_t\rangle + \frac{\eta_{x_t}^2 L_\Phi}{2}\|m_t\|^2\right]$$

On the other hand, by the definition of $m_t$, it holds that

$$\left(1-\eta_m\right)\mathbb{E}\|m_t\|^2 - \left(1-\eta_m\right)\mathbb{E}\|m_{t-1}\|^2 \leq \eta_{m_t}\langle\Delta_t, m_t\rangle - \eta_{m_t}\|m_t\|^2$$

Thus, combining the above two inequalities, it holds that

$$\mathbb{E}\left[\frac{\eta_{m_t}}{\eta_{x_t}}\Phi\left(x_t\right) + \frac{1-\eta_{m_t}}{2}\|m_t\|^2 - \frac{\eta_{m_t}}{\eta_{x_t}}\Phi\left(x_{t-1}\right) - \frac{1-\eta_{m_t}}{2}\|m_{t-1}\|^2\right]$$

$$\leq \mathbb{E}\left[-\eta_{m_t}\langle\nabla\Phi\left(x_{t-1}\right), m_t\rangle + \frac{\eta_{m_t}\eta_{x_t}L_\Phi}{2}\|m_t\| + \eta_{m_t}\langle\Delta_t, m_t\rangle - \eta_{m_t}\|m_t\|^2\right]$$

$$\leq \mathbb{E}\eta_{m_t}\langle\mathbb{E}\Delta_t - \nabla\Phi\left(x_{t-1}\right), m_t\rangle - \left(\eta_{m_t} - \frac{\eta_{m_t}\eta_{x_t}L_\Phi}{2}\right)\mathbb{E}\|m_t\|^2$$

$$\leq \frac{\eta_{m_t}}{2}\mathbb{E}\|\mathbb{E}\Delta_t - \nabla\Phi\left(x_{t-1}\right)\|^2 - \frac{\eta_{m_t} - \eta_{m_t}\eta_{x_t}L_\Phi}{2}\|m_t\|^2$$

When $\eta_{x_t} \leq \frac{1}{2L_\Phi}$, it holds that

$$\mathbb{E}\left[\frac{\eta_{m_t}}{\eta_{x_t}}\Phi\left(x_t\right) + \frac{1-\eta_{m_t}}{2}\|m_t\|^2 - \frac{\eta_{m_t}}{\eta_{x_t}}\Phi\left(x_{t-1}\right) - \frac{1-\eta_{m_t}}{2}\|m_{t-1}\|^2\right]$$

$$\leq \frac{\eta_{m_t}}{2}\mathbb{E}\|\mathbb{E}\Delta_t - \nabla\Phi\left(x_{t-1}\right)\|^2 - \frac{\eta_{m_t}}{4}\|m_t\|^2$$

$$\leq \eta_{m_t}\left(L_1 + D_z L_2\right)^2\mathbb{E}\|y_{t-1} - y^*\left(x_{t-1}\right)\|^2 + \eta_{m_t} L_1^2\left(1-\eta_z\mu\right)^{2K}\left(D_z + \frac{L_0}{\mu}\right)^2 - \frac{\eta_{m_t}}{4}\mathbb{E}\|m_t\|^2$$

$$\square$$

*Proof of Theorem 2.* By the update rules of $x_t$, $m_t$ and $y_t$, it holds that

$$\mathbb{E}\left[\frac{\eta_{m_t}}{\eta_{x_t}}\Phi\left(x_t\right) + \frac{1-\eta_{m_t}}{2}\|m_t\|^2 + \frac{4\eta_{m_t}\left(L_1 + D_z L_2\right)^2}{\mu\eta_{y_t}}\|y_t - y^*\left(x_t\right)\|^2\right.$$

$$\left. - \frac{\eta_{m_t}}{\eta_{x_t}}\Phi\left(x_{t-1}\right) - \frac{1-\eta_{m_t}}{2}\|m_{t-1}\|^2 - \frac{4\eta_{m_t}\left(L_1 + D_z L_2\right)^2}{\mu\eta_{y_t}}\|y_{t-1} - y^*\left(x_{t-1}\right)\|^2\right]$$

$$\leq \eta_{m_t}\left(L_1 + D_z L_2\right)^2\mathbb{E}\left[\|y_{t-1} - y^*\left(x_{t-1}\right)\|^2 + \eta_{m_t} L_1^2\left(1-\eta_z\mu\right)^{2K}\left(D_z + \frac{L_0}{\mu}\right)^2 - \frac{\eta_{m_t}}{4}\mathbb{E}\|m_t\|^2\right.$$

$$\left. - 2\eta_{m_t}\left(L_1 + D_z L_2\right)^2\|y_{t-1} - y^*\left(x_{t-1}\right)\|^2 + \frac{4\eta_{m_t}\left(L_1 + D_z L_2\right)^2\left(2+\mu\eta_{y_t}\right)L_1^2\eta_{x_t}^2}{\mu^2\eta_{y_t}^2}\|m_t\|^2\right]$$

$$+ \frac{8\eta_{m_t}\eta_{y_t}\left(L_1 + D_z L_2\right)^2 D_0}{\mu}.$$

When $\frac{\eta_{x_t}}{\eta_{y_t}} \leq \frac{\mu}{8L_1(L_1+D_2L_2)}$, it holds that

$$
\mathbb{E}\left[\frac{\eta_{m_t}}{\eta_{x_t}}\Phi\left(x_t\right) + \frac{1-\eta_{m_t}}{2}\|m_t\|^2 + \frac{4\eta_{m_t}\left(L_1+D_zL_2\right)^2}{\mu\eta_{y_t}}\|y_t - y^*\left(x_t\right)\|^2\right.
$$
$$
\left.-\frac{\eta_{m_t}}{\eta_{x_t}}\Phi\left(x_{t-1}\right) - \frac{1-\eta_{m_t}}{2}\|m_{t-1}\|^2 - \frac{4\eta_{m_t}\left(L_1+D_zL_2\right)^2}{\mu\eta_{y_t}}\|y_{t-1} - y^*\left(x_{t-1}\right)\|^2\right]
$$
$$
\leq -\eta_{m_t}\left(L_1+D_zL_2\right)^2\mathbb{E}\|y_{t-1}-y^*\left(x_{t-1}\right)\|^2 + \eta_{m_t}L_1^2\left(1-\eta_z\mu\right)^{2K}\left(D_z+\frac{L_0}{\mu}\right)^2 - \frac{\eta_{m_t}}{8}\mathbb{E}\|m_t\|^2
$$
$$
+ \frac{8\eta_{m_t}\eta y_t\left(L_1+D_zL_2\right)^2 D_0}{\mu}.
$$

Thus, by summing the inequality up it holds that

$$
\sum_{t=1}^{T}\frac{\eta_{m_t}}{16}\mathbb{E}\|m_t\|^2
$$
$$
\leq \frac{\eta_{m_1}}{\eta_{x_t}}\Phi\left(x_0\right) + \frac{4\eta_{m_1}\left(L_1+D_zL_2\right)^2}{\mu\eta_{y_1}}\|y_0-y^*\left(x_0\right)\|^2 - \sum_{t=1}^{T}\frac{\eta_{m_t}}{16}\mathbb{E}\|m_t\|^2
$$
$$
+ \mathbb{E}\left[\sum_{t=1}^{T-1}\left(\frac{\eta_{m_{t-1}}}{\eta_{x_{t-1}}} - \frac{\eta_{m_t}}{\eta_{x_t}}\right)\Phi\left(x_t\right) + \left(\frac{1-\eta_{m_{t-1}}}{2} - \frac{1-\eta_{m_t}}{2}\right)\|m_t\|^2\right]
$$
$$
+ \mathbb{E}\left[\left(\frac{4\eta_{m_{t-1}}\left(L_1+D_zL_2\right)^2}{\mu\eta_{y_{t-1}}} - \frac{4\eta_{m_t}\left(L_1+D_zL_2\right)^2}{\mu\eta_{y_t}}\right)\|y_0-y^*\left(x_0\right)\|^2\right]
$$
$$
- \mathbb{E}\left[\frac{\eta_{m_T}}{\eta_{x_T}}\Phi\left(x_T\right) + \frac{1-\eta_{m_T}}{2}\|m_T\|^2 + \frac{4\eta_{m_T}\left(L_1+D_zL_2\right)^2}{\mu\eta_{y_T}}\|y_T-y^*\left(x_T\right)\|^2\right] - \sum_{t=1}^{T}\frac{\eta_{m_t}}{16}\mathbb{E}\|m_t\|^2
$$
$$
- \sum_{t=1}^{T}\eta_{m_t}\left(L_1+D_zL_z\right)^2\mathbb{E}\|y_{t-1}-y^*\left(x_{t-1}\right)\|^2 + \eta_{m_t}L_1^2\left(1-\eta_z\mu\right)^{2K}\left(D_z+\frac{L_0}{\mu}\right)
$$
$$
+ \frac{8\eta_{m_t}\eta_{y_t}\left(L_1+D_zL_2\right)^2 D_0}{\mu}.
$$

When $\eta_{m_t}$ is non-increasing, $\frac{\eta_{m_t}}{\eta x_t}$ is non-increasing and $\frac{\eta_{m_t}}{\eta y_t}$ is non-increasing, it holds that

$$
\sum_{t=1}^{T}\frac{\eta_{m_t}}{16}\mathbb{E}\|m_t\|^2
$$
$$
\leq \frac{\eta_{m_1}}{\eta_{x_t}}\left(\Phi\left(x_0\right) - \underline{\Phi}\right) + \frac{4\eta_{m_1}\left(L_1+D_zL_2\right)^2}{\mu\eta_{y_1}}\|y_0-y^*\left(x_0\right)\|^2 - \sum_{t=1}^{T}\frac{\eta_{m_t}}{16}\mathbb{E}\|m_t\|^2
$$
$$
- \sum_{t=1}^{T}\eta_{m_t}\left(L_1+D_zL_z\right)^2\mathbb{E}\|y_{t-1}-y^*\left(x_{t-1}\right)\|^2 + \eta_{m_t}L_1^2\left(1-\eta_z\mu\right)^{2K}\left(D_z+\frac{L_0}{\mu}\right)
$$
$$
+ \frac{8\eta_{m_t}\eta_{y_t}\left(L_1+D_zL_2\right)^2 D_0}{\mu}.
$$

On the other hand, it holds that

$$\sum_{t=1}^{T}\mathbb{E}\eta_{m_{t+1}}\|\nabla\Phi\left(x_t\right)\|^2 \leq 2\sum_{t=1}^{T}\mathbb{E}\eta_{m_{t+1}}\|m_t\|^2 + \eta_{m_{t+1}}\|m_t - \nabla\Phi\left(x_t\right)\|^2$$

$$\leq 2\sum_{t=1}^{T}\mathbb{E}\eta_{m_t}\|m_t\|^2 + \eta_{m_{t+1}}\|m_t - \nabla\Phi\left(x_t\right)\|^2$$

$$\leq \frac{32\eta_{m_1}}{\eta_{x_t}}\left(\Phi\left(x_0\right) - \underline{\Phi}\right) + \frac{128\eta_{m_1}\left(L_1 + D_z L_2\right)^2}{\mu\eta_{y_1}}\|y_0 - y^*\left(x_0\right)\|^2 - \sum_{t=1}^{T}\mathbb{E}\|m_t\|^2 + \frac{256\eta_{m_t}\eta_{y_t}\left(L_1 + D_z L_2\right)^2 D_0}{\mu}$$

$$- 32\sum_{t=1}^{T}\eta_{m_t}\left(L_1 + D_z L_z\right)^2\mathbb{E}\|y_{t-1} - y^*\left(x_{t-1}\right)\|^2 + 16\eta_{m_t}L_1^2\left(1 - \eta_z\mu\right)^{2K}\left(D_z + \frac{L_0}{\mu}\right)$$

$$+ 2\mathbb{E}\|\nabla\Phi\left(x_0\right)\|^2 + \sum_{t=1}^{T}4\eta_{m_t}\mathbb{E}\|\mathbb{E}\Delta_t - \nabla\Phi\left(x_{t-1}\right)\|^2 + 4\eta_{x_t}^2/\eta_{m_t}L_1^2\|m_t\|^2 + 2\eta_{m_t}^2\mathbb{E}\|\Delta_t - \mathbb{E}\Delta_t\|^2$$

$$\leq \frac{32\eta_{m_1}}{\eta_{x_t}}\left(\Phi\left(x_0\right) - \underline{\Phi}\right) + \frac{128\eta_{m_1}\left(L_1 + D_z L_2\right)^2}{\mu\eta_{y_1}}\|y_0 - y^*\left(x_0\right)\|^2 + \frac{256\eta_{m_t}\eta_{y_t}\left(L_1 + D_z L_2\right)^2 D_0}{\mu}$$

$$- 32\sum_{t=1}^{T}\eta_{m_t}\left(L_1 + D_z L_z\right)^2\mathbb{E}\|y_{t-1} - y^*\left(x_{t-1}\right)\|^2 + 16\eta_{m_t}L_1^2\left(1 - \eta_z\mu\right)^{2K}\left(D_z + \frac{L_0}{\mu}\right)$$

$$+ 2\mathbb{E}\|\nabla\Phi\left(x_0\right)\|^2 + \sum_{t=1}^{T}\left(4\eta_{x_t}^2/\eta_{m_t}L_1^2 - 1\right)\|m_t\|^2$$

$$+ \sum_{t=1}^{T}4\eta_{m_t}\left(2\left(L_1 + D_z L_2\right)^2\|y_{t-1} - y^*\left(x_{t-1}\right)\|^2 + 2L_1^2\left(1 - \eta_z\mu\right)^{2K}\left(D_z + \frac{L_0}{\mu}\right)^2\right)$$

$$+ \sum_{t=1}^{T}2\eta_{m_t}^2\left(L_0^2 + 2L_1^2\left(KL_1^2 D_z^2 + 2K\eta_z^2 L_1^2 L_0^2 + \frac{2KL_0^2}{\mu^2}\right) + 2D_z^2 L_1^2\right)$$

$$\leq \frac{32\eta_{m_1}}{\eta_{x_t}}\left(\Phi\left(x_0\right) - \underline{\Phi}\right) + \frac{128\eta_{m_1}\left(L_1 + D_z L_2\right)^2}{\mu\eta_{y_1}}\|y_0 - y^*\left(x_0\right)\|^2 + 2\mathbb{E}\|\nabla\Phi\left(x_0\right)\|^2$$

$$+ \sum_{t=1}^{T}\frac{256\eta_{m_t}\eta_{y_t}\left(L_1 + D_z L_2\right)^2 D_0}{\mu} + 8\eta_{m_t}L_1^2\left(1 - \eta_z\mu\right)^{2K}\left(D_z + \frac{L_0}{\mu}\right)^2 + 16\eta_{m_t}L_1^2\left(1 - \eta_z\mu\right)^{2K}\left(D_z + \frac{L_0}{\mu}\right)$$

$$+ \sum_{t=1}^{T}2\eta_{m_t}^2\left(L_0^2 + 2L_1^2\left(KL_1^2 D_z^2 + 2K\eta_z^2 L_1^2 L_0^2\frac{2KL_0^2}{\mu^2}\right) + 2D_z^2 L_1^2\right)$$

Thus, when K is large $\left(1 - \eta_z\right)^{2K}$ will small, and it holds that

$$\min_{t\in\{1,\cdots,T\}}\mathbb{E}\|\nabla\Phi\left(x_t\right)\|^2 = \mathcal{O}\left(\frac{1 + \sum_{t=1}^{T}\eta_{m_t}\eta_{y_t} + \eta_{m_t}^2}{\sum_{t=1}^{T}\eta_{m_t}}\right)$$

$\square$

## C  PROOF OF COROLLARY 2

*Proof of Corollary 2.* According to Theorem 2, when taking $\eta_{x_t} = \Theta(1/t)$, $\eta_{y_t} = \Theta(1/t)$ and $\eta_{m_t} = \Theta(1/t)$, we get $\min_{t\in\{1,\cdots,T\}}\mathbb{E}\|\nabla\Phi(x_t)\|^2 = \mathcal{O}(1/\log T)$. Thus, to achieve $\min_{t\in\{1,\cdots,T\}}\mathbb{E}\|\nabla\Phi(x_t)\|^2 \leq \epsilon$, we get $T = \Omega(e^{1/\epsilon})$.

Meanwhile, according to Corollary 1, $\epsilon_{stab} = \mathcal{O}(T^q/n)$.

We have $\log\epsilon_{stab} = \mathcal{O}(1/\epsilon)$. $\square$

## D   PROOF OF COROLLARY 3

*Proof of Corollary 3.* We directly formulate the optimization problem as follows:

$$\min_{\eta_m, T} \left\{ (1 + \alpha \eta_m)^T, \qquad s.t. \ \frac{\beta}{T \eta_m} + \gamma \eta_m \le \epsilon \right\},$$

For fixed $T$, we can solve the optimal $\eta_m^* = \frac{\epsilon T - \sqrt{\epsilon^2 T^2 - 4\gamma \beta T}}{2\gamma T}$ when $T \ge \frac{4\gamma\beta}{\epsilon}$.

Then the problem becomes

$$\min_{\eta_m, T} \left\{ \left(1 + \alpha \frac{\epsilon T - \sqrt{\epsilon^2 T^2 - 4\gamma\beta T}}{2\gamma T}\right)^T, \qquad s.t. \ T \ge \frac{4\gamma\beta}{\epsilon} \right\},$$

By taking the derivative, the function value is decreasing while T increases. Thus, the optimal value is

$$\lim_{T \to \infty} \left(1 + \alpha \frac{\epsilon T - \sqrt{\epsilon^2 T^2 - 4\gamma\beta T}}{2\gamma T}\right)^T = e^{\frac{\alpha}{\epsilon\gamma}}$$

Thus, we obtain that the optimal value is in order of $e^{\mathcal{O}(1/\epsilon)}$. □

## E   ADDITIONAL EXPERIMENTS

### E.1   ABLATION STUDY ON LARGE $K$

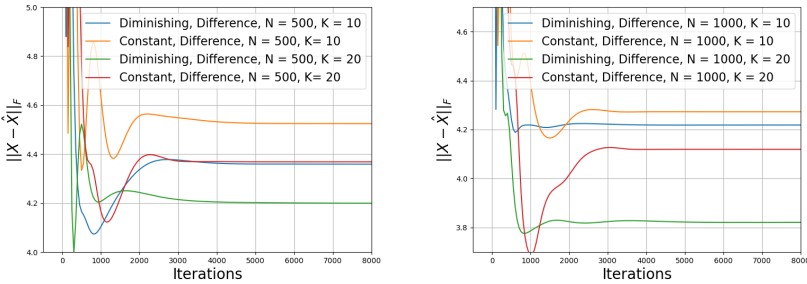

Figure 3: Results for Toy Example. The left figure shows the results when $N = 500$, and the right figure shows the results when $N = 1000$.

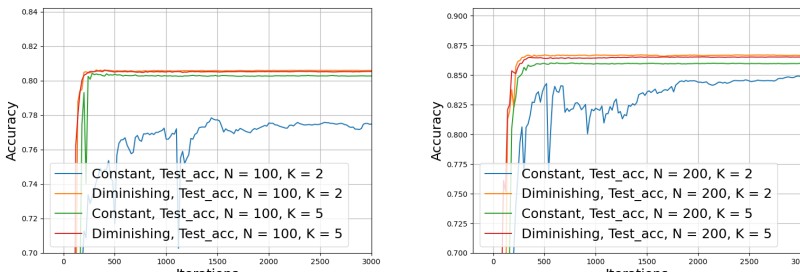

Figure 4: Results for MNIST classification. The left figure shows the results when $n = 100$, and the right figure shows the results when $n = 200$.

### E.1.1   RESULTS FOR SECTION 5.1

For the toy example, we increase $K$ from 10 to 20, and the results are shown in Figure 3. It is shown in the figure that, increasing $K$ can help achieve smaller error, while the diminishing step size achieves a better performance no matter if $K$ is large or small.

### E.1.2 RESULTS FOR SECTION 5.2

For the MNIST example, we increase $K$ from 2 to 5. The results are shown in Figure 4. Similar to the results in the toy example, increasing $K$ can help achieve smaller errors, while the diminishing step size achieves a better performance no matter if $K$ is large or small.

### E.2 ADDITIONAL EXPERIMENTS ON CIFAR10

With the same setting in Section 5.2, we change the dataset from MNIST to CIFAR10 (Krizhevsky et al., 2009) and change the number of samples in the training set from 5000 to 20000. The results are shown in Figure 5. Because of the representation ability of Lenet-5, the results of constant stepsize and the diminishing stepsize are quite similar. However, at the end of 5000 iterations of training, the diminishing stepsize achieves a little higher accuracy for both $n = 100$ case and $n = 200$ case.

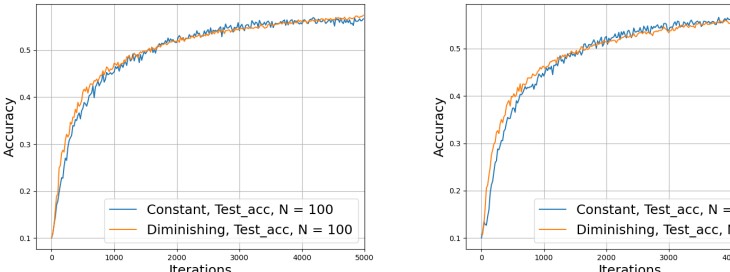

Figure 5: Results for CIFAR10 classification. The left figure shows the results when $n = 100$, and the right figure shows the results when $n = 200$.

