# OpenReview forum: "Exploring the Generalization Capabilities of AID-based Bi-level Optimization"
_ICLR.cc/2024/Conference — Submitted to ICLR 2024_

### Official Review · Reviewer_FTaq · 2023-10-28

**Soundness:** 2 fair
**Presentation:** 3 good
**Contribution:** 3 good
**Rating:** 3
**Confidence:** 3

**Summary:**

This paper studies optimization algorithms for the approximate implicit differentiation (AID) based bilevel optimization problems, which own an inner and an outer objective function. Existing studies mainly consider the development of optimization algorithms and convergence analysis, and there seems merely single work on the algorithmic stability-based generalization assessment [1] as described in this paper. Instead, the paper considers the argument-level uniformly stability and generalization analysis for the complex AID algorithms where the upper level problem is non-convex. Especially, there are several intermediate variables in the so-called AID algorithms, which may bring challenges to the analysis.
	Including the convergence analysis, this paper further gives a similar result on algorithmic stability bounds as compared to existing bilevel work [1] and single-level one [2]. However, this may not obviously highlight the advantages to [1] or contributions of this work to the community.

[1] Stability and generalization of bilevel programming in hyperparameter optimization, NeurIPS’21
[2] Bilevel Optimization: Convergence Analysis and Enhanced Design, ICML’21

**Strengths:**

(1) This paper considers the AID based optimization algorithm, which is commonly used in bilevel problems. There are several intermediate variables in the so-called AID algorithms, which may bring challenges to the analysis.
(2) And the AID algorithm indeed is more complex than classical iterative differentiation (ITD) algorithms. Furthermore, the convergence analysis of the AID algorithm (Algorithm 1) is also provided. The generalization and convergence analysis provide some theoretical guarantees to the practice.
(3) The settings of step sizes for updating these variables seems practically feasible, which may further provide guidance to the realistic practice to improve and balance the convergence rate and generalization performance.
(4) This paper is fairly well-written, the representation of the learning algorithms (like the commonly used decomposition strategy) is helpful for readers to better understand the learning objects.

**Weaknesses:**

I checked parts of the proof in the appendix, and not check all the proof. There may exist some typos and technical flaws in the analysis that I have checked as follows:

(1) Even though the algorithm is complex with several intermediate variables involved in the analysis, the proof techniques for each are analogous to [1] and some existing stability-based work [3-5]. And the employed argument-based stability tool indeed has been used previously [4].
	The authors are suggested to further add some descriptions of the employed assumptions (e.g., the e-accuracy from [2], Assumption 3 from …), existing stability-based work and provide some discussions on the differences with them. Besides, it is kindly suggested to highlight the challenges and difficulties of the proof, so as to help the readers understand the contributions of this work.

[1] Stability and generalization of bilevel programming in hyperparameter optimization, NeurIPS’21
[2] Bilevel Optimization: Convergence Analysis and Enhanced Design, ICML’21
[3] Stability and generalization, JMLR’02
[4] Algorithmic stability and hypothesis complexity, ICML’17


(2) As described in the contribution, I’d like to know from which perspective of this paper, the robustness of the AID algorithm can be demonstrated. And the so-called robustness is the adversary robustness or some statistical robustness [1] ?

[1] Huber P J. Robust statistics.

(3) In Definition 1, only perturbation in the validation (or outer) dataset is considered. Why not consider perturbation in the inner-level training dataset? Some discussions of the reasons for this definition are needed.
	In fact, there are prior works that consider perturbation on both datasets for a special case of bilevel learning, meta learning. See [1].

[1] Generalization of Model-Agnostic Meta-Learning Algorithms: Recurring and Unseen Tasks, NeurIPS 2021.

(4) However, Algorithms 1&2 are not the same as the previous works that I’ve read [1-3]. It is understandable that approximation algorithms are not unique for AID algorithms (see Table 1 in [2]). Please indicate in detail which related works has proposed or used these algorithms?

[1] Bilevel Optimization: Convergence Analysis and Enhanced Design
[2] Optimizing Millions of Hyperparameters by Implicit Differentiation
[3] Approximation Methods for Bilevel Programming


(5) Although the authors give an example of non-convex upper problem, it could be still interesting and necessary to further analyze other conditions. For some examples in convergence analysis, an asymptotic analysis are conduct under the assumptions that the lower and upper functions are convex and strongly convex [1,2]. [3] studied the setting where average version \Fai() is strongly convex or convex, and g(x,.) is strongly convex.
It would be better if this issue is considered, but not necessarily required.

[1] A generic first-order algorithmic framework for bi-level programming beyond lower-level singleton
[2] Improved bilevel model: Fast and optimal algorithm with theoretical guarantee
[3] Approximation methods for bilevel programming.

(6) In Corollary 1, the constants \alpha and \beta are involved in the learning rate without detailed definition? Is the \beta here the same as the stability definition in Definition 1 or 2?

(7) In the Appendix A, the conclusion of Lemma 8 seems to be wrongly used in following lemmas and Theorem 1. Dz indeed could only bound z^K, which does not hold for z^0 or z^k where k<K.
	This could be the most serious weakness and will render some arguments in the paper invalid.

(8) In the proof of Lemma 11 (see Appendix, P.14), the rules for updating variable ‘m’ seems contractive with the Algorithm 1. Besides, the mixture usage of y and y*(x) indeed cause confusion (see Lemma 16 and Theorem 2).

The following are the potential typos or mistakes that have been found:
	In Remark 3, the \Theta -> O
	In the proof of Lemma 8 (P.12), A is undefined
	In the proof of Lemma 9 (P.12), symbol ‘+’ before \eta_z is ignored; (P.13 Line.3) symbol ‘=’
	In the proof of Lemma 10 (P.13), symbol ‘<=’ is ignored; the description of “By selecting \eta_y such that \mu \eta_y ≥ L_2 \eta_y” seems wrong; (P.14) x should lie in [0,0.5] in the first line; ‘2’-> ‘K+1’ in the 4-th line.
	In the proof of Lemma 11 (P.14), the rules for updating m seems contractive with the Algorithm 1.
	In the proof of Lemma 15 (P.18), ‘X’ or ‘x’?
	In the proof of Lemma 16 (P.18), ‘\nabla_t’ -> ‘\nabla_y’; single ‘\nabla’ -> ‘\nabla^2_yy’

The authors are kindly asked to check these issues when conducting the revision.

**Questions:**

Please see Weaknesses stated as above.

---

> ### Author Response · Authors · 2023-11-22
> **Response to Reviewer Reviewer FTaq(1/2)**
>
> (1) Even though the algorithm is complex with several intermediate variables involved in the analysis, the proof techniques for each are analogous to [1] and some existing stability-based work [3-5]. And the employed argument-based stability tool indeed has been used previously [4]. The authors are suggested to further add some descriptions of the employed assumptions (e.g., the e-accuracy from [2], Assumption 3 from …), existing stability-based work and provide some discussions on the differences with them. Besides, it is kindly suggested to highlight the challenges and difficulties of the proof, so as to help the readers understand the contributions of this work.[1] Stability and generalization of bilevel programming in hyperparameter optimization, NeurIPS’21 [2] Bilevel Optimization: Convergence Analysis and Enhanced Design, ICML’21 [3] Stability and generalization, JMLR’02 [4] Algorithmic stability and hypothesis complexity, ICML’17
>
> > First, our stability analysis is far different from [1], where [1] reduces the bi-level problem into a single-level problem. More specifically, in [1], $y_t$ can be written as a function of $x_t$, i.e. $y_t = F(x_t)$. By analyzing the property of function $F$, the problem can be reduced to a single-level problem. However, as $y_t$ depends on all previous iterations in the AID-based algorithm, we use a very different way, by making the connection of successive iterates on all $x,y$ and $m$.
> >  We do borrow some proof from single-level stability analysis from [3-4]. However, we have to deal with the case from a single variable to 3 correlated variables, where they use different update rules and even based on different functions.
>
>
> (2) As described in the contribution, I’d like to know from which perspective of this paper, the robustness of the AID algorithm can be demonstrated. And the so-called robustness is the adversary robustness or some statistical robustness [1] ?[1] Huber P J. Robust statistics.
> >  Sorry for the confusion, we use robustness to show more stable on changing one data sample in the dataset. Since it will cause confusion, we deleted the word in the contribution.
>
> (3) In Definition 1, only perturbation in the validation (or outer) dataset is considered. Why not consider perturbation in the inner-level training dataset? Some discussions of the reasons for this definition are needed. In fact, there are prior works that consider perturbation on both datasets for a special case of bilevel learning, meta learning. See [1].[1] Generalization of Model-Agnostic Meta-Learning Algorithms: Recurring and Unseen Tasks, NeurIPS 2021.
> > As we discuss in Section 3.2, usually we have much more data in the lower-level problem than the upper-level problem. It is important to first consider the bottleneck, the upper-level problem. [1] gives an example of how the 'inner loop' will affect the difference of iterates, but to consider the inner loop, we have to first define a proper decomposition rule like Section 3.2. The most difficult part is that (III) in Section 3.2 may not be non-positive when changing the constraints for constrained optimization. Thus, we decided to analyze the generalization of the inner level to future work.

---

> ### Author Response · Authors · 2023-11-22
> **Response to Reviewer Reviewer FTaq(2/2)**
>
> (4) However, Algorithms 1&2 are not the same as the previous works that I’ve read [1-3]. It is understandable that approximation algorithms are not unique for AID algorithms (see Table 1 in [2]). Please indicate in detail which related works has proposed or used these algorithms?[1] Bilevel Optimization: Convergence Analysis and Enhanced Design [2] Optimizing Millions of Hyperparameters by Implicit Differentiation [3] Approximation Methods for Bilevel Programming
>
> > For AID-based algorithm, there are different ways to approximate $\nabla_{yy} g(x,y)^{-1}\nabla_y f(x,y)$. In algorithm 1, we use gradient descent to approximate it. In Table 1 in [2], our algorithm belongs to CG row instead of taking conjugate gradient descent, we use gradient descent for simplicity.
>
> (5) Although the authors give an example of non-convex upper problem, it could be still interesting and necessary to further analyze other conditions. For some examples in convergence analysis, an asymptotic analysis are conduct under the assumptions that the lower and upper functions are convex and strongly convex [1,2]. [3] studied the setting where average version \Fai() is strongly convex or convex, and g(x,.) is strongly convex. It would be better if this issue is considered, but not necessarily required.[1] A generic first-order algorithmic framework for bi-level programming beyond lower-level singleton [2] Improved bilevel model: Fast and optimal algorithm with theoretical guarantee [3] Approximation methods for bilevel programming.
> > Thank you for the suggestion. The difficulty of the analysis is that the property on $\Phi$ only controls the function property of $f(x,y)$ when y is near to $y*(x)$. However, the stability analysis needs to hold on arbitrary (x,y). Thus, the condition that $\Phi$ is strongly convex is not enough to give a better stability bound. We will give some more conditions for strongly convex cases in future work.
>
>
>
> (6) In Corollary 1, the constants \alpha and \beta are involved in the learning rate without detailed definition? Is the \beta here the same as the stability definition in Definition 1 or 2?
> > The $\alpha$ and $\beta$ are two constants in the learning rate, which will affect the value of $q = \frac{2C_m\alpha + L_1\beta}{2C_m\alpha + L_1\beta+1}$ in Corollary 1.
>
>
> (7) In the Appendix A, the conclusion of Lemma 8 seems to be wrongly used in following lemmas and Theorem 1. Dz indeed could only bound z^K, which does not hold for z^0 or z^k where k<K. This could be the most serious weakness and will render some arguments in the paper invalid.
> > Thanks for pointing it out, we have updated Lemma 8. Now, it holds for all k that $||z_t^k|| \leq D_z :=||z_0|| + \frac{L_0}{\mu}$. Since $\|z_0\|$ and $\frac{L_0}{\mu}$ are constants, it will not affect the latter proof.
>
>
> (8) In the proof of Lemma 11 (see Appendix, P.14), the rules for updating variable ‘m’ seems contractive with the Algorithm 1. Besides, the mixture usage of y and y*(x) indeed cause confusion (see Lemma 16 and Theorem 2).
> > Sorry for the typos in the algorithm, we have updated Algorithm 1.
>
>
> The following are the potential typos or mistakes that have been found: In Remark 3, the \Theta -> O In the proof of Lemma 8 (P.12), A is undefined In the proof of Lemma 9 (P.12), symbol ‘+’ before \eta_z is ignored; (P.13 Line.3) symbol ‘=’ In the proof of Lemma 10 (P.13), symbol ‘<=’ is ignored; the description of “By selecting \eta_y such that \mu \eta_y ≥ L_2 \eta_y” seems wrong; (P.14) x should lie in [0,0.5] in the first line; ‘2’-> ‘K+1’ in the 4-th line. In the proof of Lemma 11 (P.14), the rules for updating m seems contractive with the Algorithm 1. In the proof of Lemma 15 (P.18), ‘X’ or ‘x’? In the proof of Lemma 16 (P.18), ‘\nabla_t’ -> ‘\nabla_y’; single ‘\nabla’ -> ‘\nabla^2_yy’The authors are kindly asked to check these issues when conducting the revision.
> > Thanks for you pointing them out, we have updated the paper accordingly.

---

### Official Review · Reviewer_JTQs · 2023-10-30

**Soundness:** 3 good
**Presentation:** 3 good
**Contribution:** 3 good
**Rating:** 6
**Confidence:** 3

**Summary:**

The paper studies bilevel optimization problems. It aims at understanding the generalization behavior AID method.  They provide excess uniform stability bounds on validation data under certain assumptions, the stability bound has a scaling of $O(T^q/n)$.

**Strengths:**

1. The main strength of the paper is that it investigates the stability of the AIM method in Bilevel optimization.
2. The paper is well-written and theoretically solid.

**Weaknesses:**

1. The influence of the inner iteration K is not discussed. In Theorem 2, the paper only gives 'K is large enough'. However, in the experiments, the author only uses K=2. I think this is confumsing. Is there any relation between K and T? [1] proposed a method where K=1, so may be 'K is large enough' is not needed.
2. If K is needed to be large, i think the performance with different K should be given.
[1] Dagréou M, Ablin P, Vaiter S, et al. A framework for bilevel optimization that enables stochastic and global variance reduction algorithms[J]. Advances in Neural Information Processing Systems, 2022, 35: 26698-26710.

**Questions:**

1. The influence of the inner iteration K is not discussed. In Theorem 2, the paper only gives 'K is large enough'. However, in the experiments, the author only uses K=2. I think this is confumsing. Is there any relation between K and T? [1] proposed a method where K=1, so may be 'K is large enough' is not needed.
2. If K is needed to be large, i think the performance with different K should be given.
[1] Dagréou M, Ablin P, Vaiter S, et al. A framework for bilevel optimization that enables stochastic and global variance reduction algorithms[J]. Advances in Neural Information Processing Systems, 2022, 35: 26698-26710.

---

> ### Author Response · Authors · 2023-11-22
> **Response to Reviewer Reviewer JTQs**
>
> The influence of the inner iteration K is not discussed. In Theorem 2, the paper only gives 'K is large enough'. However, in the experiments, the author only uses K=2. I think this is confumsing. Is there any relation between K and T? [1] proposed a method where K=1, so may be 'K is large enough' is not needed.  If K is needed to be large, i think the performance with different K should be given.
>
> [1] Dagréou M, Ablin P, Vaiter S, et al. A framework for bilevel optimization that enables stochastic and global variance reduction algorithms[J]. Advances in Neural Information Processing Systems, 2022, 35: 26698-26710.
>
> > In Therorem 2, we have added the theorectical results for $K = \Theta(\log T)$, and we increase the K in both experiments. The results are shown in Appendix, the results indicate that increasing K can help get smaller error, while the diminishing learning rate is indeed better than constant learning rate even K is large.
> > [1] gives the results when K = 1, because they assume higher order smothness like $\nabla^3 g$ is Lipschitz smooth and $\nabla^2 f$ is Lipschitz smooth. With weaker condition, we can only show the K with a large value.

---

### Official Review · Reviewer_CjiH · 2023-10-30

**Soundness:** 3 good
**Presentation:** 3 good
**Contribution:** 3 good
**Rating:** 8
**Confidence:** 3

**Summary:**

The paper first presents a novel analysis framework for examining multi-level variables within the stability of bi-level optimization. It provides a stability analysis for non-convex optimization with various learning rate configurations. The paper offers convergence results for AID-based methods and highlights the trade-off between convergence and stability of these algorithms. The authors conduct an ablation study of the parameters and assess the performance of these methods on real-world tasks, providing practical guidance on managing and minimizing gaps in bi-level optimization. The experimental results corroborate the theoretical findings, demonstrating the effectiveness and potential applications of AID-based bi-level optimization methods.

**Strengths:**

(1) The authors first present an innovative analysis framework aimed at systematically examining the behavior of multi-level variables within the stability of bi-level optimization, which derives theoretical stability bounds for AID-based bi-level methods.

(2) This study illustrates the uniform stability of AID-based techniques, even under mild conditions. The stability bounds observed are comparable to those encountered in nonconvex single-level optimization and ITD-based bi-level methods.

(3) Their research reveals generalization gap results related to optimization errors, offering a deeper understanding of the trade-offs between convergence and stability of the AID-based bi-level optimization, ultimately improving the eﬀiciency and effectiveness of the AID-based methods.

**Weaknesses:**

(1) This analysis in this paper appears heavily reliant on smoothness assumptions for the inner and outer functions, the theoretical results may be not suited to loss function with L1 penalty. Therefore, the results in this paper do not yet seem generalizable to the broader field of artificial intelligence.

(2) The paper does not seem to consider the generalization ability of the inner problem, but sometimes the solution to the inner problem is equally important. Therefore, it would be better if the generalization ability of the lower problem could also be analyzed.

(3) The paper considers the trade-off between convergence and stability of AID based algorithms, but only gives the cases of constant learning rate and diminishing learning rate, without discussing the impact of decaying learning rate at different rates to choose better learning rate.

**Questions:**

(1) The paper has discussed the relation between generalization error and learning rate, could other critical factors, such as batch size, impact the generation capabilities of the proposed AID algorithms?

(2) Might it be feasible to incorporate more challenging datasets and tasks in order to validate the theoretical analysis, such as CIFAR?

(3) Will there be more significant challenges in extending the conclusions of this paper to AID-based bi-level optimization variants?

---

> ### Author Response · Authors · 2023-11-22
> **Response to Reviewer Reviewer CjiH**
>
> Weakness:
> (1) This analysis in this paper appears heavily reliant on smoothness assumptions for the inner and outer functions, the theoretical results may be not suited to loss function with L1 penalty. Therefore, the results in this paper do not yet seem generalizable to the broader field of artificial intelligence.
> > Thank you for pointing it out. Since as far as we know, this is the first paper that analyzes the generalization of AID-based bilevel algorithms, we start from the simple case where the function has nice smooth properties. We have added the suggestion in the future work in conclusion that there is still a mystery for the proper choice of stepsize, and for the weaker conditions of the upper-level and lower-level objective function, we will leave for future work.
>
>
> (2) The paper does not seem to consider the generalization ability of the inner problem, but sometimes the solution to the inner problem is equally important. Therefore, it would be better if the generalization ability of the lower problem could also be analyzed.
> > As the inner problem will have a much larger dataset than the upper level, it is more important to analyze the upper level first. After that, we can further analyze the inner-level problem. Different from the upper-level objective, the inner-level objective is considered as a constraint, such the generalization needs to be defined first on the constraints for constrained optimization problems. We will consider it in future work.
>
>
> (3) The paper considers the trade-off between convergence and stability of AID based algorithms, but only gives the cases of constant learning rate and diminishing learning rate, without discussing the impact of decaying learning rate at different rates to choose better learning rate.
> > For stability results and convergence results, the formulations are complicated, where statbility is in order of $O(\sum_{t=1}^T \Pi_{k=t+1}^T (1+ \eta_{x_k}\eta_{m_k}+\eta_{m_k} + \eta_{y_k} L_1)(1+\eta_{x_t})\eta_{m_t}/n)$ and convergence is in order of $\mathcal{O}(\frac{1+\sum_{k=1}^T {\eta_{y_k}\eta_{m_k} + \eta_{m_k}^2}}{\sum_{k=1}^T\eta_{m_k}})$. Thus, we just analyze some popular learning rate choices instead of getting an optimal schedule. We will leave it to future work.
>
> Questions:
> (1) The paper has discussed the relation between generalization error and learning rate, could other critical factors, such as batch size, impact the generation capabilities of the proposed AID algorithms?
> > We don't think batch size will impact the stability of the algorithm. In principle, when we increase the batch size, the probability of choosing the different sample will increase, while the weight for each sample to update the model will decrease. Based on simple calculations, we can find the effective weight for the different sample is still 1/n. Thus, it will impact the analysis in the same way as batch size equals 1. Further, Hardt et al. (2016) show that the stability will remain the same stability bound.
> > However, it will impact the convergence part by reducing variance, which will be clearly discussed in future work.
>
> (2) Might it be feasible to incorporate more challenging datasets and tasks in order to validate the theoretical analysis, such as CIFAR?
> > Due to the limited time, we added an experiment in the Appendix. We use dataset CIFAR10 and the rest settings are the same as MNIST. The results show that with diminishing step size, the algorithm can perform better than constant step size.
>
>
> (3) Will there be more significant challenges in extending the conclusions of this paper to AID-based bi-level optimization variants?
> > We are sorry that we get confused about the AID-based bi-level optimization variants. However, we think for any small modification of the algorithm, the whole proof needs to be updated with at least 2-3 lemmas.

---

### Official Review · Reviewer_R6J2 · 2023-11-04

**Soundness:** 2 fair
**Presentation:** 2 fair
**Contribution:** 2 fair
**Rating:** 3
**Confidence:** 3

**Summary:**

The authors investigate the stability of a specific double-loop AID-based algorithm for bilevel optimization. They demonstrate that, under specific conditions, this algorithm achieves $O(T^q/n)$-stable, which is of a similar order as ITD-based methods. The authors also conduct a convergence analysis under specific conditions of stepsizes. By combining the stability and convergence results, they determine the generalization ability of the proposed algorithm. Their experimental findings show that the generalization ability for diminishing learning rates outperforms the generalization ability for constant rates.

**Strengths:**

S1. The exploration of stability and generalization in bi-level optimization is an under-explored area, and the topic addressed in this work is both interesting and significant.

S2. A novel analytical framework for examining the stability of bilevel optimization has been developed.

S3. The paper is easy to follow.

**Weaknesses:**

W1. I didn't examine all the proofs of the main results in depth, but I did identify some errors in certain proofs, including the proof of Theorem 1 (specifically, details in Q3 and minor comments below). It would be advisable for the authors to review and validate all their results for accuracy.

W2. The paper doesn't adequately cover closely related papers on AID-based bi-level optimization algorithms, including:

[1] K. Ji, J. Yang, and Y. Liang. ``Bilevel Optimization: Convergence Analysis and Enhanced Design.” ICML 2021.

[2] M. Dagreou et al. ``A framework for bilevel optimization that enables stochastic and global variance reduction algorithms.” NeurIPS 2022.

W3. The convergence analysis in this paper lacks novelty, as the approach presented in Chen et al. (2022) can be readily extended from constant learning rates to time-evolving learning rates in a standard manner.

W4. There are numerous unnecessary typos, details in Minor Comments below.

**Questions:**

Q1. Why do the authors focus on the specific double-loop AID-based bi-level optimization algorithm in Algorithm 1? There are several single-loop AID-based bi-level optimization algorithms available, as demonstrated in Dagréou et al.'s work ``A framework for bilevel optimization that enables stochastic and global variance reduction algorithms" in NeurIPS (2022), as well as the related references.

Q2. Why are the solutions of bi-level optimization problems in Section 3.2 unique, considering the definitions of $(\bar{x}, \bar{y})$ and $(x^*, y^*)$? Note that Remark 1 mentions that the bi-level optimization problem is likely to have a nonconvex objective with respect to $x$.

Q3. Why can we use the same sample for different datasets? Note that there is an internal randomness of algorithm $\mathcal{A}$ (i.e., Algorithm 1). The proof of Theorem 1 should be more rigorous. The proof of Theorem 2.2 in Hardt et al. (2016) may provide valuable insights.

Minor Comments:

(1)The sample spaces for $\xi_i$ and $\zeta_i$ can differ; for instance, refer to the toy example in Section 5.1.

(2)In Algorithm 2, where the sample $\zeta_t^{(K+1)}$ is used?

(3)There are numerous unnecessary typos, such as:

(i)the long inequality of generalization error on page 4: $A$ should be $\mathcal{A}$, $N$ should be $n$, and $\mathbb{E}_{z, D_v}$ should be $\mathbb{E}_{z}$ in (IV).

(ii)line 2 before Proposition 2: $x_i$ should be $\xi_i$.

(iii)Proposition 2: $\mathbb{E}_{\mathcal{A}, D_z}$ should be $\mathbb{E}_{\mathcal{A}, D_v}$.

(iv)Definition 2: check and correct the statement such as clarify the meaning of
$D_v$ and $D_v’$, and so on.

(v)In the beginning of Section 4.5, what follows the word ``However,”?

(vi)After the toy example in Section 5.1, swap the positions of $A_1$ and $A_2$.

(vii)Proof of Lemma 8 on Page 12: $\frac{L_0}{\eta_z}$ should be $L_0 \eta_z$. There are also other unnecessary typos in the proofs on Lemma 9, Corollary 1 and elsewhere.

(4)Where is the proof of Proposition 3? According to Definition 2, it's not immediately obvious, as there are different samples in $f$, and the assumptions on $f$ are made for the same sample.

(5)Equation (2) in Theorem 2: $L_{\Phi}$ is not defined before, so clarify its meaning.

(6)How large is $K$ in Theorem 2? Does it have any impact on the experiments?

(7)Where is the proof of Corollary 2? At the end of the proof of Corollary 3, how does $e^{\frac{\alpha}{\epsilon \gamma}}$ become $\mathcal{O}(e^{1/\epsilon})$? Is this statement correct?

(8)Toy example in Section 5.1: Why is $(\hat{X}, \hat{y})$ considered the ground truth? Please note that there is an $L_2$ regularization term in the context.

(9)Does the lower-level objective in Section 5.2 is strongly convex for $y$?

---

> ### Author Response · Authors · 2023-11-22
> **Response to Reviewer Reviewer R6J2(2/2)**
>
> (5) Equation (2) in Theorem 2: $L_\Phi$ is not defined before, so clarify its meaning.
> > Similar to the gradient descent, we have a necessary condition to ensure descent that the step size is less than or equal to $1/L$, where $L$ is the gradient Lipschitz constant of the function. Here, we prove that the gradient of function $\Phi$ is Lipschitz with constant $L_\Phi$. We have updated the value of $L_\Phi$ in Thereom 2.
>
> (6) How large $K$ is  in Theorem 2? Does it have any impact on the experiments?
> > According to the proof of Theorem $K = \Theta(\log T)$, which we updated it in Thereom 2. We add an additional experiment by increasing K in the Appendix. It turns out that by increasing $K$, we can get a better generalization error, but the conclusion that diminishing learning rate works better than the constant learning rate still holds.
>
> (7) Where is the proof of Corollary 2? At the end of the proof of Corollary 3, how does $e ^{\alpha/(\gamma\epsilon)}$ become $O(e ^{1/\epsilon})$? Is this statement correct?
> > We have updated the statement in Corollary 2 and Corollary 3. Instead of giving the order of $\epsilon _{stab}$, we give the order of $\log \epsilon _{stab}$. The proof of Corollary 2 has been added to the Appendix. It is correct that $\log e ^{\frac{\alpha}{\gamma\epsilon}}$ becomes $O(1/\epsilon)$.
>
> (8)Toy example in Section 5.1: Why is $(\hat{X},\hat{y})$ considered the ground truth? Please note that there is an $L_2$ regularization term in the context.
> > As the goal is to recover $\hat{X}$ and $\hat{y}$, we consider $(\hat{X}, \hat{y}))$ as the ground truth. $L_2$ regularization in the lower-level problem is for generalization as a prior because of limited data. It is a well-known trick to improve generalization which helps $y$ close to $\hat{y}$. $L_2$ regularization in the upper-level problem is for softening the constraint that $X$ is orthogonal, which helps $X$ close $\hat{X}$. Thus, $L_2$ regularization helps to find the ground truth.
>
> (9)Does the lower-level objective in Section 5.2 strongly convex for $y$?
> > No, the lower-level objective in Section 5.2 is not strongly convex for $y$. The experiment is to show that even without the strong convexity in the lower-level objective, our conclusion still holds.

---

> ### Author Response · Authors · 2023-11-22
> **Response to Reviewer Reviewer R6J2(1/2)**
>
> 1. Why do the authors focus on the specific double-loop AID-based bi-level optimization algorithm in Algorithm 1? There are several single-loop AID-based bi-level optimization algorithms available, as demonstrated in Dagréou et al.'s work ``A framework for bilevel optimization that enables stochastic and global variance reduction algorithms" in NeurIPS (2022), as well as the related references.
> > Different from ``A framework for bilevel optimization that enables stochastic and global variance reduction algorithms", we do not assume the Lipschitz continuity of $\nabla^2_x f(x,y)$ and $\nabla^3_y g(x,y)$. With weaker assumptions, it is hard to prove the convergence of a single-loop algorithm.  We have added the discussion to the related work that  Dagréou et al. (2022)  show the convergence under higher-order smoothness setting.
> 2. Why are the solutions of bi-level optimization problems in Section 3.2 unique, considering the definitions of $(\bar{x}, \bar{y})$ and $(x^*,y^*)$? Note that Remark 1 mentions that the bi-level optimization problem is likely to have a nonconvex objective with respect to x.
> > Sorry for this confusion, we define $(\bar{x},\bar{y})$ and $(x^*,y^*)$ as one of the optimal solution, where we change $=$ into $\in$. Because any optimal solution in the set shares the same function value, the decomposition will remain the same in Section 3.2.
> 3.  Why can we use the same sample for different datasets? Note that there is an internal randomness of algorithm $\mathcal{A}$((i.e., Algorithm 1). The proof of Theorem 1 should be more rigorous. The proof of Theorem 2.2 in Hardt et al. (2016) may provide valuable insights.
> > Based on the definition, we want to bound $\mathbb{E}_\mathcal{A} ||\mathcal{A}(D_t,D_v) - \mathcal{A}(D_t,D_v')||$, where for each instance of algorithm $\mathcal{A}$, two trajectories share the same randomness. In practice, the randomness of $\mathcal{A}$ is the sample index. Thus, two trajectories either sample the same data point or sample the only different data point in the dataset. This technique is identical to the proof of Hardt et al.(2016).
>
> Minor Comments:(1)(2)(3)
> > Thank you for pointing them out, we have updated them in the revision.
>
> (4) Where is the proof of Proposition 3? According to Definition 2, it's not immediately obvious, as there are different samples in $f$, and the assumptions on $f$ are made for the same sample.
> > We have updated Definition 2, it is a typo to use $D_v$ and $D_v'$ in f. The correct version is $|| E _{\mathcal{A}, D _v \sim P _{D _v} ^n, D' _v \sim P _{D _v}^n}[ f(\mathcal{A}(D _t,D _v),z) -  f(\mathcal{A}(D _t,D _v'),z)]||\leq\beta,\ \forall D _t \in \mathcal{Z} _t^q, z \in Z _v.$ Based on the new definition 2, it is obvious that Proposition 3 is correct.

---

### Meta-Review · Area_Chair_KAx9 · 2023-12-21

**Metareview:**

This paper demonstrates the uniform stability of approximate implicit differentiation (AID)-based methods in bilevel optimization, even when the outer-level function is nonconvex. It yields outcomes comparable to solving a single-level nonconvex problem. Experimental results align with the theoretical findings, showcasing the effectiveness and potential applications of the AID-based methods. The reviewers have identified several issues about this paper : (1) technical errors/flaws in the proof; (2) lack of novelty in the analysis; (3) lack of comparison with highly related papers. The author rebuttal does not fully address these issues. I agree with the reviewers’ evaluation and thus recommend rejection.

**Justification For Why Not Higher Score:**

technical errors/flaws in the proof.

**Justification For Why Not Lower Score:**

N/A

---

### Decision · Program_Chairs · 2024-01-16

Reject